# Monocular Dynamic View Synthesis: A Reality Check

**Hang Gao**[1,†]    **Ruilong Li**[1]    **Shubham Tulsiani**[2]    **Bryan Russell**[3]    **Angjoo Kanazawa**[1]

[1]UC Berkeley        [2]Carnegie Mellon University        [3]Adobe Research

## Abstract

We study the recent progress on dynamic view synthesis (DVS) from monocular video. Though existing approaches have demonstrated impressive results, we show a discrepancy between the practical capture process and the existing experimental protocols, which effectively leaks in multi-view signals during training. We define *effective multi-view factors* (EMFs) to quantify the amount of multi-view signal present in the input capture sequence based on the relative camera-scene motion. We introduce two new metrics: co-visibility masked image metrics and correspondence accuracy, which overcome the issue in existing protocols. We also propose a new iPhone dataset that includes more diverse real-life deformation sequences. Using our proposed experimental protocol, we show that the state-of-the-art approaches observe a 1-2 dB drop in masked PSNR in the absence of multi-view cues and 4-5 dB drop when modeling complex motion. Code and data can be found at http://hangg7.com/dycheck.

## 1   Introduction

Dynamic scenes are ubiquitous in our everyday lives – people moving around, cats purring, and trees swaying in the wind. The ability to capture 3D dynamic sequences in a "casual" manner, particularly through monocular videos taken by a smartphone in an uncontrolled environment, will be a cornerstone in scaling up 3D content creation, performance capture, and augmented reality.

Recent works have shown promising results in dynamic view synthesis (DVS) from a monocular video [1, 2, 3, 4, 5, 6, 7, 8]. However, upon close inspection, we found that there is a discrepancy between the problem statement and the experimental protocol employed. As illustrated in Figure 1, the input data to these algorithms either contain frames that "teleport" between multiple camera viewpoints at consecutive time steps, which is impractical to capture from a single camera, or depict quasi-static scenes, which do not represent real-life dynamics.

In this paper, we provide a systematic means of characterizing the aforementioned discrepancy and propose a better set of practices for model fitting and evaluation. Concretely, we introduce *effective multi-view factors* (EMFs) to quantify the amount of multi-view signal in a monocular sequence based on the relative camera-scene motion. With EMFs, we show that the current experimental protocols operate under an effectively multi-view regime. For example, our analysis reveals that the aforementioned practice of camera teleportation makes the existing capture setup akin to an Olympic runner taking a video of a moving scene without introducing any motion blur.

The reason behind the existing experimental protocol is that monocular DVS is a challenging problem that is also hard to evaluate. Unlike static novel-view synthesis where one may simply evaluate on held-out views of the captured scene, in the dynamic case, since the scene changes over time, evaluation requires another camera that observes the scene from a different viewpoint at the same time. However, this means that the test views often contain regions that were never observed in the input sequence. Camera teleportation, *i.e.*, constructing a temporal sequence by alternating samples

---

† Work partially done as part of HG's internship at Adobe.

36th Conference on Neural Information Processing Systems (NeurIPS 2022).

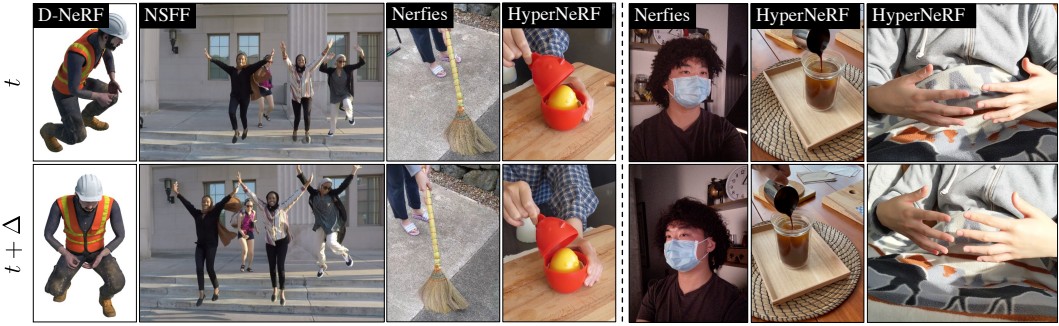

(a) Teleporting camera motion ($\Delta = 0.2$s)          (b) Quasi-static scene motion ($\Delta = 2$s)

Figure 1: **Visualizing the training data of existing benchmarks.** Existing datasets operate under the effective multi-view regime: The sequences either have (a) teleporting camera motion or (b) quasi-static scene motion. These motions leak multi-view cues, *e.g.*, the model can observe the human (1st column) and hands (last column) at roughly the same pose from different viewpoints.

from different cameras, addresses this issue at the expense of introducing multi-view cues, which are unavailable in the practical single-camera capture.

We propose two sets of metrics to overcome this challenge without the use of camera teleportation. The first metric enables evaluating only on pixels that were seen in the input sequence by computing the co-visibility of every test pixel. The proposed co-visibility mask can be used to compute masked image metrics (PSNR, SSIM [9] and LPIPS [10]). While the masked image metrics measure the quality of rendering, they do not directly measure the quality of the inferred scene deformation. Thus, we also propose a second metric that evaluates the quality of established point correspondences by the percentage of correctly transferred keypoints (PCK-T) [11]. The correspondences may be evaluated between the input and test frames or even within the input frames, which enable evaluation on sequences that are captured with only a single camera.

We conduct extensive evaluation on existing datasets [5, 7] as well as a new dataset that includes more challenging motion and diverse scenes. When tested on existing datasets without camera teleportation, the state-of-the-art methods observe a 1-2 dB drop in masked PSNR and ~5% drop in PCK-T. When tested on complex motion with the proposed dataset, existing approaches observe another 4-5 dB drop in masked PSNR and ~30% drop in PCK-T, suggesting a large room for improvement. We encourage future works to report EMFs on new data and adopt our experimental protocol to evaluate monocular DVS methods. Code and data are available at our project page.

## 2 Related work

**Non-rigid structure from motion (NR-SfM).** Traditional NR-SfM tackles the task of dynamic 3D inference by fitting parametric 3D morphable models [12, 13, 14, 15, 16, 17, 18, 19], or fusing non-parametric depth scans of generic dynamic scenes [20, 21, 22, 23, 24]. All of these approaches aim to recover accurate surface geometry at each time step and their performance is measured with ground truth 3D geometry or 2D correspondences with PCK [25] when such ground truth is not available. In this paper, we analyze recent dynamic view synthesis methods whose goal is to generate a photo-realistic novel view. Due to their goal, these methods do not focus on evaluation against ground truth 3D geometry, but we take inspiration from prior NR-SfM works to evaluate the quality of the inferred 3D dynamic representation based on correspondences. We also draw inspiration from previous NR-SfM work that analyzed camera/object speed and 3D reconstruction quality [26, 27].

**Monocular dynamic neural radiance fields (dynamic NeRFs).** Dynamic NeRFs reconstruct moving scenes from multi-view inputs or given pre-defined deformation template [28, 29, 30, 31, 32, 33, 34, 35]. In contrast, there is a series of recent works that seek to synthesize high-quality novel views of generic dynamic scenes given a monocular video [1, 2, 3, 4, 5, 6, 7, 8]. These works can be classified into two categories: a deformed scene is directly modeled as a time-varying NeRF in the world space [1, 4, 6] or as a NeRF in canonical space with a time-dependent deformation [2, 3, 5, 7, 8]. The evaluation protocol in these works inherit from the original static-scene NeRF [36] that quantify the rendering quality of held-out viewpoints using image metrics, *e.g.*, PSNR. However, in dynamic

scenes, PSNR from an unseen camera view may not be meaningful since the novel view may include regions that were never seen in the training view (unless the method can infer unseen regions using learning based approaches). Existing approaches resolve this issue by incorporating views from multiple cameras during training, which we show results in an effectively multi-view setup. We introduce metrics to measure the difficulties of an input sequence, a monocular dataset with new evaluation protocol and metrics, which show that existing methods have a large room for improvement.

# 3 Effective multi-view in a monocular video

We consider the problem of dynamic view synthesis (DVS) from a monocular video. A monocular dynamic capture consists of a single camera observing a moving scene. The lack of simultaneous multi-view in the monocular video makes this problem more challenging compared to the multi-view setting, such as reconstructing moving people from multiple cameras [30, 34, 37].

Contrary to the conventional perception that the effect of multi-view is binary for a capture (single versus multiple cameras), we show that it can be characterized on a continuous spectrum. Our insight is that a monocular sequence contains *effective* multi-view cues when the camera moves much faster than the scene, though technically the underlying scene is observed only once at each time step.

## 3.1 Characterizing effective multi-view in a monocular video

Although a monocular video only sees the scene from one viewpoint at a time, depending on the capture method, it can still contains cues that are effectively similar to those captured by a multi-view camera rig, which we call as effective multi-view. As shown in Figure 2, when the scene moves significantly slower than the camera (to the far right end of the axis), the same scene is observed from multiple views, resulting in multi-view capture. In this case, DVS re-

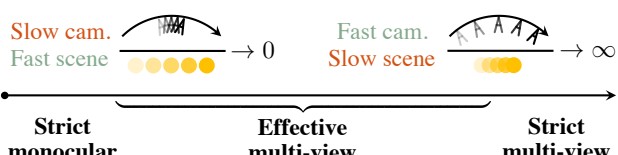

Figure 2: **The spectrum of effective multi-view in a monocular video.** A video captured by a single camera can still have multi-view cues when the camera moves much faster than the scene. We disentangle recent advances in DVS given a monocular video from such phenomena.

duces to a well-constrained multi-view stereo problem at each time step. Consider another case where the camera moves significantly faster compared to the scene so that it observes roughly the same scene from different viewpoints. As the camera motion approaches the infinity this again reduces the monocular capture to a multi-view setup. We therefore propose to characterize the amount of multi-view cues by the relative camera-scene motion.

## 3.2 Quantifying effective multi-view in a monocular video

For practicality, we propose two metrics, referred to as effective multi-view factors (EMFs). The first metric, full EMF $\Omega$ is defined as the relative ratio between the motion magnitude of the camera to the scene, which in theory characterizes the effective multi-view perfectly, but in practice can be expensive and challenging to compute. The second metric, angular EMF $\omega$ is defined as the camera angular velocity around the scene look-at point, which only considers the camera motion; while approximate, it is easy to compute and characterizes object-centric captures well.

**Full EMF $\Omega$: ratio of camera-scene motion magnitude.** Consider a monocular video of a moving scene over a set of time steps $\mathcal{T}$. At each discrete time $t \in \mathcal{T}$, let the camera's 3D location be $\mathbf{o}_t$. We consider each point $\mathbf{x}_t$ on the domain of observed scene surface $\mathbb{S}_t^2 \subset \mathbb{R}^3$. We define the camera-scene motion as the expected relative ratio,

$$\Omega = \mathop{\mathbb{E}}_{t,t+1 \in \mathcal{T}} \left[ \mathop{\mathbb{E}}_{\mathbf{x}_t \in \mathbb{S}_t^2} \left[ \frac{\|\mathbf{o}_{t+1} - \mathbf{o}_t\|}{\|\mathbf{x}_{t+1} - \mathbf{x}_t\|} \right] \right], \tag{1}$$

where the denominator $\mathbf{x}_{t+1} - \mathbf{x}_t$ denotes the 3D scene flow and the the numerator $\mathbf{o}_{t+1} - \mathbf{o}_t$ denotes the 3D camera motion, both over one time step forward. The 3D scene flow can be estimated via the

| | #Train cam. | Duration | FPS | Depth | Kpt. | Sequences |
|---|---|---|---|---|---|---|
| D-NeRF [3] | ~150 | $1-3$s | 60 | - | - | 8 MV |
| HyperNeRF [7] | 2 | $8-15$s | 15 | - | - | 3 MV + 13 SV |
| Nerfies [5] | 2 | $8-15$s | 5/15 | - | - | 4 MV |
| NSFF [4] | 24 | $1-3$s | 15/30 | Estimated [39] | - | 8 MV |
| iPhone (proposed) | 1 | $8-15$s | 30/60 | Lidar | ✓ | 7 MV + 7 SV |

Table 1: **Summary of the existing and proposed iPhone datasets.** Existing datasets operate under the effective multi-view regime, teleporting between multiple cameras during training to generate a synthetic monocular video. Unlike previous protocols, the proposed iPhone dataset consists of dynamic sequences captured by a *single* smoothly moving camera. It also has accompanying depth maps for training supervision and labeled keypoints for evaluation. "MV" denotes multi-camera capture, and "SV" denotes single-camera capture.

2D dense optical flow field and the metric depth map when available, or monocular depth map from off-the-shelf approaches [38, 39] in the general case. Please see the Appendix for more details. Note that $\Omega$ in theory captures the effective multi-view factor for any sequence. However, in practice, 3D scene flow estimation is an actively studied problem and may suffer from noisy or costly predictions.

**Angular EMF $\omega$: camera angular velocity.** We introduce a second metric $\omega$ that is easy to compute in practice. We make an additional assumption that the capture has a single look-at point in world space, which often holds true, particularly for captures involving a single centered subject. Specifically, given a look-at point $\mathbf{a}$ by triangulating the optical-axes of all cameras (as per [5]) and the frame rate $N$, the camera angular velocity $\omega$ is computed as a scaled expectation,

$$\omega = \mathop{\mathbb{E}}_{t,t+1 \in \mathcal{T}} \left[ \arccos \left( \frac{\langle \mathbf{a} - \mathbf{o}_t, \mathbf{a} - \mathbf{o}_{t+1} \rangle}{\|\mathbf{a} - \mathbf{o}_t\| \cdot \|\mathbf{a} - \mathbf{o}_{t+1}\|} \right) \right] \cdot N. \tag{2}$$

Note that even though $\omega$ only considers the camera motion, it is indicative of effective multi-view in the majority of existing captures, which we describe in Section 4.1.

For both $\Omega$ and $\omega$, the larger the value, the more multi-view cue the sequence contains. For future works introducing new input sequences, we recommend always reporting angular EMF for its simplicity and reporting full EMF when possible. Next we inspect the existing experimentation practices under the lens of effective multi-view.

## 4 Towards better experimentation practice

In this section, we reflect on the existing datasets and find that they operate under the effective multi-view regime, with either teleporting camera motion or quasi-static scene motion. The reason behind the existing protocol is that monocular DVS is challenging from both the modeling and evaluation perspective. While the former challenge is well known, the latter is less studied, as we expand below. To overcome the existing challenge in the evaluation and enable future research to experiment with casually captured monocular video, we propose a better toolkit, including two new metrics and a new dataset of complex motion in everyday lives.

### 4.1 Closer look at existing datasets

We investigate the datasets used for evaluation in D-NeRF [3], HyperNeRF [7], Nerfies [5], and NSFF [4]. Table 1 shows their statistics. We evaluate the amount of effective multi-view cues via the proposed EMFs, shown in Figure 3. We find that existing datasets have large EMF values on both metrics. For example, the HyperNeRF dataset has an $\omega$ as large as ~$200°$/s. To put these numbers in context, a person imaging an object 3m away has to move at $1$m/s to get an $\omega = 20°$/s (close to the statistics in the proposed dataset). Some datasets exhibit $\omega$ higher than $120°$/s, which is equivalent to a camera motion faster than the Olympic 100m sprint record, without incurring any motion blur.

Visualizing the actual training data shown in Figure 1 reveals that existing datasets feature non-practical captures of either (1) teleporting/fast camera motion or (2) quasi-static/slow scene motion. The former is not representative of practical captures from a hand-held camera, *e.g.*, a smartphone, while the latter is not representative of moving objects in daily life. Note that, out

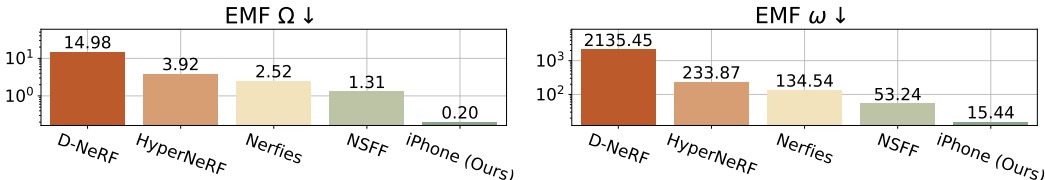

Figure 3: **Statistics of effective multi-view factors (EMFs) across different datasets.** Existing datasets have high EMFs, indicating the abundance of multi-view cues during training (note that the y-axis is in log scale). The proposed iPhone dataset features a single camera, capturing the moving scene with a smooth motion, and thus has smaller EMFs. Our results help ground the difficulty of each dataset for DVS from a monocular video.

of the 23 multi-camera sequences that these prior works used for quantitative evaluation, 22 have teleporting camera motion, and 1 has quasi-static scene motion – the CURLS sequence shown at the $5^{th}$ column in Figure 1. All 13 single-camera sequences from HyperNeRF [7] used for qualitative evaluation have quasi-static scene motion.

The four datasets also share a similar data protocol for generating effective multi-view input sequences from the original multi-camera rig capture. In Figure 4, we illustrate the teleporting protocol used in Nerfies [5] and HyperNeRF [7] as a canonical example. They sample alternating frames from two physical cameras (left and right in this case) mounted on a rig to create the training data. NSFF [4] samples alternating frames from 24 cameras based on the data released from Yoon *et al.* [40]. D-NeRF [3] experiments on synthetic dynamic scenes where cameras

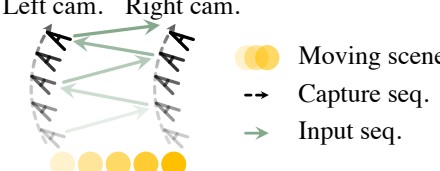

Figure 4: **Camera teleportation from a multi-camera rig.** As an example, we show the data protocol used in Nerfies [5] and HyperNeRF [7].

are randomly placed on a fixed hemisphere at every time step, in effect teleporting between 100-200 cameras. We encourage you to visit our project page to view the input videos from these datasets.

Existing works adopt effective multi-view capture for two reasons. First, it makes monocular DVS more tractable. Second, it enables evaluating novel view on the full image, without worrying about the visibility of each test pixel, as all camera views were visible during training. We show this effect in Figure 5. When trained with camera teleportation ($3^{rd}$ column), the model can generate a high-quality full image from the test view. However, when trained without camera teleportation ($4^{th}$ column), the model struggles to hallucinate unseen pixels since NeRFs [5, 36] are not designed to predict completely unseen portions of the scene, unless they are specifically trained for generalization [41]. Next, we propose new metrics that enable evaluation without using camera teleportation. Note that when the model is trained without camera teleportation, the rendering quality also degrades, which we also evaluate.

### 4.2 Our proposed metrics

While the existing setup allows evaluating on the full rendered image from the test view, the performance under such evaluation protocol, particularly with teleportation, confounds the efficacy of the proposed approaches and the multi-view signal present in the input sequence. To evaluate with an actual monocular setup, we propose two new metrics that evaluate only on seen pixels and measure the correspondence accuracy of the predicted deformation.

**Co-visibility masked image metrics.** Existing works evaluate DVS models with image metrics on the *full* image, *e.g.*, PSNR, SSIM [9] and LPIPS [10], following novel-view synthesis evaluation on static scenes [36]. However, in dynamic scenes, particularly for monocular capture with multi-camera validation, the test view contains regions that may not have been observed at all by the training camera. To circumvent this issue without resorting to camera teleportation, for each pixel in the test image, we propose *co-visibility* masking, which tests how many times a test pixel has been observed in the training images. Specifically, we use optical flow to compute correspondences between every test image and the training images, and only keep test pixels that have enough correspondences in the training images via thresholding. This results in a mask, illustrated in Figure 5, which we use to confine the image metrics. We follow the common practice from the image generation literature and adopt masked metrics, mPSNR and mLPIPS [42, 43]. Note that NSFF [4] adopts similar metrics

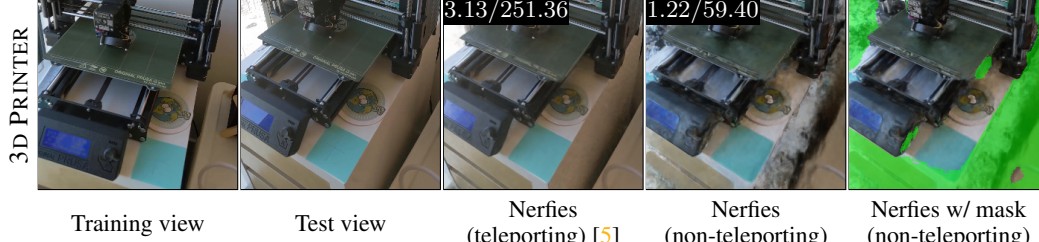

Figure 5: **Evaluation without teleportation with the co-visibility mask.** $\Omega/\omega$ metrics of the input sequence are annotated on the top-left. Existing works avoid evaluating on unseen pixels by camera teleportation ($3^{rd}$ column). Naively training with the non-teleporting (smooth) camera trajectory causes evaluation issues on the full image ($4^{th}$ column), since a NeRF model cannot hallucinate unseen regions. We propose to only evaluate seen regions during training by a co-visibility mask (we show non-visible regions in green at the last column). Note that the model performs poorly on seen regions as well when trained without camera teleportation.

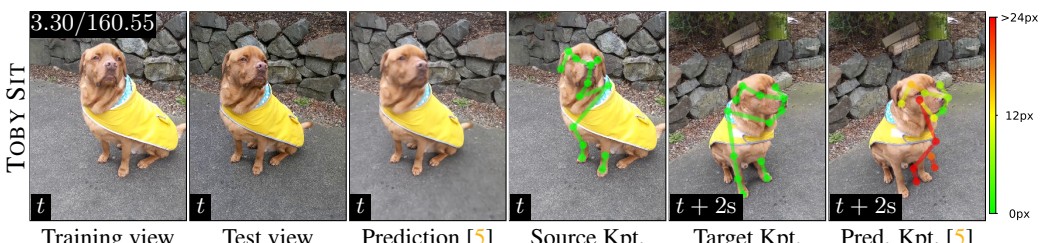

Figure 6: **High-quality novel-view synthesis does not imply accurate correspondence modeling.** $\Omega/\omega$ metrics of the input sequence are shown on the top-left. The time steps of the ground-truth data and predictions are shown on the bottom-left. Using Nerfies [5] as an example, we show that the model renders high-quality results ($3^{rd}$ column) without modeling accurate correspondences (last column). Transferred keypoints are colorized by a heatmap of end-point error, overlaid on the ground-truth target frame.

but for evaluating the rendering quality on foreground versus background regions. We additionally report mSSIM by partial convolution [44], which only considers seen regions during its computation. More details are in the Appendix. Using masked image metrics, we quantify the performance gap in rendering when a model is trained with or without multi-view cues in Section 5.1.

**Percentage of correctly transferred keypoints (PCK-T).** Correspondences lie at the heart of traditional non-rigid reconstruction [21], which is overlooked in the current image-based evaluation. We propose to evaluate 2D correspondences across training frames with the percentage of correctly transferred keypoints (PCK-T) [11], which directly evaluates the quality of the inferred deformation. Specifically, we sparsely annotate 2D keypoints across input frames to ensure that each keypoint is fully observed during training. For correspondence readout from existing methods, we use either root finding [45] or scene flow chaining. Please see the Appendix for details on our keypoint annotation, correspondence readout, and metric computation. As shown in Figure 6, evaluating correspondences reveal that high quality image rendering does not necessarily result in accurate correspondences, which indicates issues in the underlying surface, due to the ambiguous nature of the problem.

### 4.3 Proposed iPhone dataset

Existing datasets can be rectified by removing camera teleportation and evaluated using the proposed metrics, as we do in Section 5.1. However, even after removing camera teleportation, the existing datasets are still not representative of practical in-the-wild capture. First, the existing datasets are limited in motion diversity. Second, the evaluation baseline in existing datasets is small, which can hide issues in incorrect deformation and resulting geometry. For these reasons, we propose a new dataset called the *iPhone dataset* shown in Figure 7. In contrast to existing datasets with repetitive object motion, we collect 14 sequences featuring non-repetitive motion, from various categories such as generic objects, humans, and pets. We deploy three cameras for multi-camera capture – one hand-held moving camera for training and two static cameras of large baseline for evaluation. Furthermore, our iPhone dataset comes with metric depth from the lidar sensors, which we use to provide ground-truth depth for supervision. In Section 5.2, we show that depth supervision, together

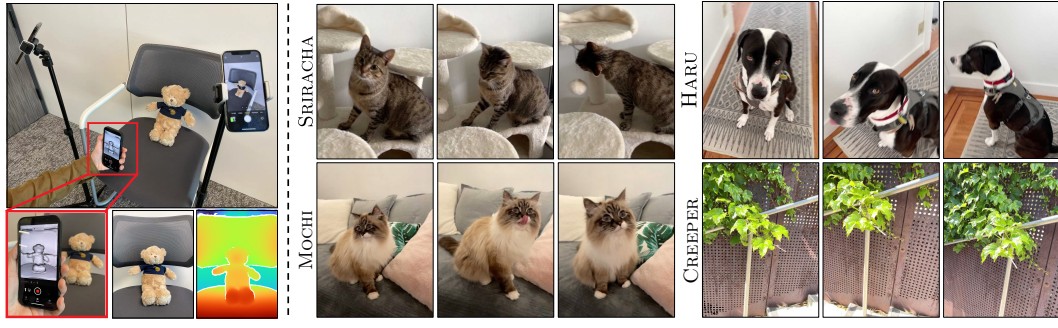

| (a) Capture setup | (b) Sampled sequences |

Figure 7: **Capture setup and sampled sequences from our proposed iPhone dataset.** The proposed iPhone dataset has a single hand-held camera for training and multiple cameras with a large baseline for validation. It has accompanying depth from the iPhone sensor and features diverse and complex real-life motions.

with other regularizations, is beneficial for training DVS models. Please see the Appendix for details on our multi-camera capture setup, data processing, and more visualizations.

## 5    Reality check: re-evaluating the state of the art

In this section, we conduct a series of empirical studies to disentangle the recent progress in dynamic view synthesis (DVS) given a monocular video from effective multi-view in the training data. We evaluate current state-of-the-art methods when the effective multi-view factor (EMF) is low.

**Existing approaches and baselines.**    We consider the following state-of-the-art approaches for our empirical studies: NSFF [4], Nerfies [5] and HyperNeRF [7]. We choose them as canonical examples for other approaches [1, 2, 3, 6, 8, 34, 35], discussed in Section 2. We also evaluate time-conditioned NeRF (T-NeRF) as a common baseline [1, 3, 4]. Unlike the state-of-the-art methods, it is not possible to extract correspondences from a T-NeRF. A summary of these methods can be found in the Appendix.

**Datasets.**    We evaluate on the existing datasets as well as the proposed dataset. For existing datasets, we use the multi-camera captures accompanying Nerfies [5] and HyperNeRF [7] for evaluation. Due to their similar capture protocol, we consider them as a single dataset in our experiment (denoted as the Nerfies-HyperNeRF dataset). It consists of 7 sequences in total, which we augment with keypoint annotations. Our dataset has 7 multi-camera captures and 7 single-camera captures. We evaluate novel-view synthesis on the multi-camera captures and correspondence on all captures. Our data adopts the data format from the Nerfies-HyperNeRF dataset, with additional support for depth and correspondence labels. All videos are at $480p$ resolution and all dynamic scenes are inward-facing.

**Masked image and correspondence metrics.**    Following Section 4.2, we evaluate co-visibility masked image metrics and the correspondence metric. We report masked image metrics: mPSNR, mSSIM [9, 44], and mLPIPS [4, 10, 42, 43]. We visualize the rendering results with the co-visibility mask. For the correspondence metric, we report the percentage of correctly transferred keypoints (PCK-T) [11] with threshold ratio $\alpha = 0.05$. Additional visualizations of full image rendering and inferred correspondences can be found in the Appendix.

**Implementation details.**    We consolidate Nerfies [5] and HyperNeRF [7] in one codebase using JAX [46]. Compared to the original official code releases, our implementation aligns all training and evaluation details between models and allows correspondence readout. Our implementation reproduces the quantitative results in the original papers. We implement T-NeRF in the same codebase. For NSFF [4], we tried both the official code base [47] and a public third-party re-implementation [48], where the former fails to converge on our proposed iPhone dataset while the latter works well. We thus report results using the third-party re-implementation. However, note that both the original and the third-party implementation represent the dynamic scene in normalized device coordinates (NDC). As NDC is designed for forward-facing but not considered inward-facing scenes, layered artifacts may appear due to its log-scale sampling rate in the world space, as shown in Figure 9. More details

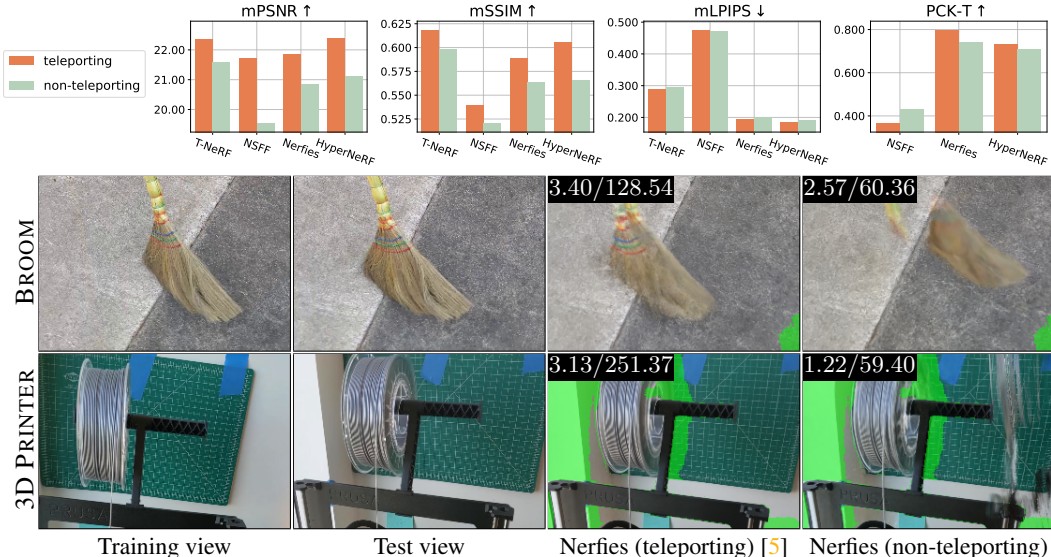

Figure 8: **Impact of effective multi-view on the Nerfies-HyperNeRF dataset.** $\Omega/\omega$ metrics of the input sequence are shown on the top-left. We compare the existing camera teleporting setting and our non-teleporting setting. (Top): Quantitative results of different models using our proposed evaluation metrics. (Bottom): Qualitative comparison using Nerfies as an example. Two settings use the same set of co-visibility masks computed from common training images. Visualizations of other models are in the Appendix.

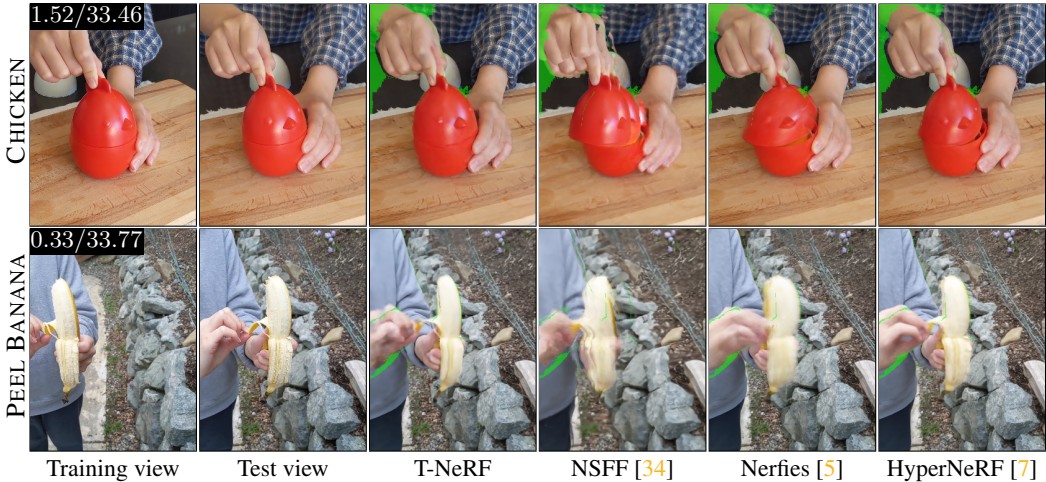

Figure 9: **Qualitative results on the Nerfies-HyperNeRF dataset without camera teleportation.** $\Omega/\omega$ metrics of the input sequence are shown on the top-left. Existing approaches struggle at modeling dynamic regions.

about aligning the training procedure and remaining differences are provided in the Appendix. Code, pretrained models, and data are available on the project page.

## 5.1 Reality check on the Nerfies-HyperNeRF dataset

**Impact of effective multi-view.** We first study the impact of effective multi-view on the Nerfies-HyperNeRF [5, 7] dataset. In this experiment, we rectify the effective multi-view sequences by only using the left camera during training as opposed to both the left and right cameras, illustrated in Figure 4. We denote the original setting as "teleporting" and the rectified sequences as "non-teleporting". We train all approaches under these two settings with the same held-out validation frames and same set of co-visibility masks computed from common training frames. In Figure 8 (Top), all methods perform better across all metrics when trained under the teleporting setting compared to the non-teleporting one, with the exception of PCK-T for NSFF. We conjecture that this is because that NSFF has additional optical flow supervision, which is more accurate without

| $\Omega = 1.30$ $\omega = 51.53$ | mPSNR↑ | mSSIM↑ | mLPIPS↓ | PCK-T↑ |
|---|---|---|---|---|
| T-NeRF | **21.55** | **0.595** | 0.297 | - |
| NSFF [4] | 19.53 | 0.521 | 0.471 | 0.422 |
| Nerfies [5] | 20.85 | 0.562 | 0.200 | 0.756 |
| HyperNeRF [7] | 21.16 | 0.565 | **0.192** | **0.764** |

Table 2: **Benchmark results on the rectified Nerfies-HyperNeRF dataset.** Please see the Appendix for the breakdown over 7 multi-camera sequences.

| $\Omega = 0.24$ $\omega = 15.18$ | mPSNR↑ | mSSIM↑ | mLPIPS↓ | PCK-T↑ |
|---|---|---|---|---|
| T-NeRF | **16.96** | **0.577** | 0.379 | - |
| NSFF [4] | 15.46 | 0.551 | 0.396 | 0.256 |
| Nerfies [5] | 16.45 | 0.570 | 0.339 | **0.453** |
| HyperNeRF [7] | 16.81 | 0.569 | **0.332** | 0.400 |

Table 3: **Benchmark results on the proposed iPhone dataset.** Please see the Appendix for the breakdown over 7 multi-camera sequences of complex motion.

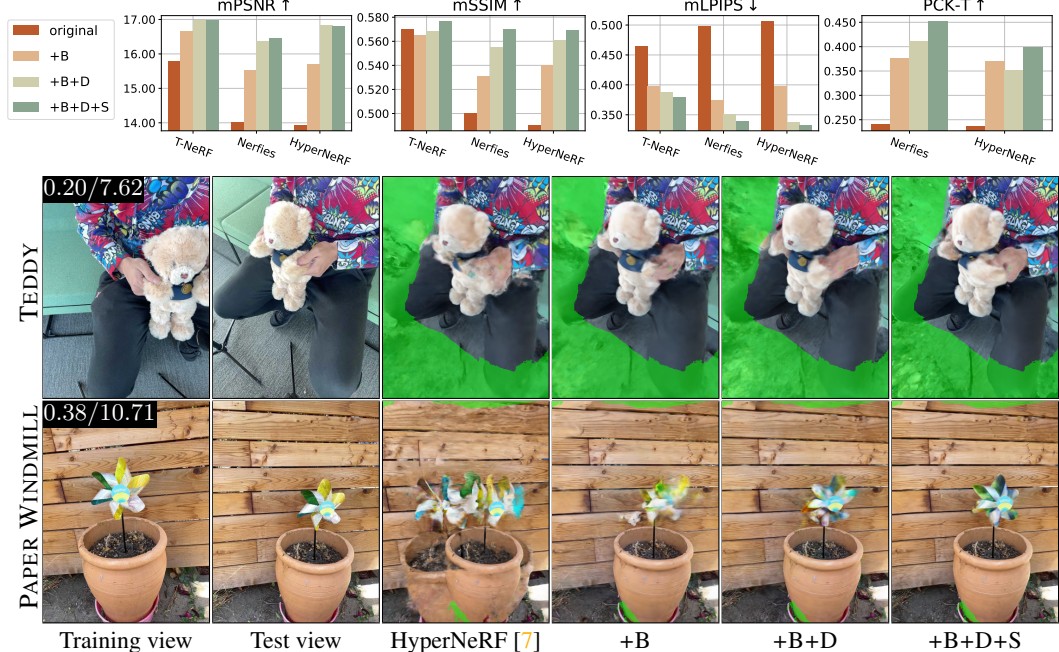

Figure 10: **Ablation study on improving the state of the art on the proposed iPhone dataset.** $\Omega/\omega$ metrics of the input sequence are shown on the top-left. +B, +D, +S denotes random background compositing [32], additional metric depth supervision [1, 4] from iPhone sensor, and surface sparsity regularizer [49], respectively.

camera teleportation. In Figure 8 (Bottom), we show qualitative results using Nerfies (we include visualizations of the other methods in the Appendix). Without effective multi-view, Nerfies fails at modeling physically plausible shape for broom and wires. Our results show that the effective multi-view in the existing experimental protocol inflates the synthesis quality of prior methods, and that truly monocular captures are more challenging.

**Benchmark results without camera teleportation.** In Table 2 and Figure 9, we report the quantitative and qualitative results under the non-teleporting setting. Note that our implementation of the T-NeRF baseline performs the best among all four evaluated models in terms of mPSNR and mSSIM. In Figure 9, we confirm this result since T-NeRF renders high-quality novel view for both sequences. HyperNeRF produces the most photorealistic renderings, measured by mLPIPS. However it also produces distorted artifacts that do not align well with the ground truth (*e.g.*, the incorrect shape in the CHICKEN sequence).

## 5.2 Reality check on the proposed iPhone dataset

**Ablation study on improving the state of the art.** We find that existing methods perform poorly out-of-the-box on the proposed iPhone dataset with more diverse and complex real-life motions. In Figure 10 (Bottom), we demonstrate this finding with HyperNeRF [7] for it achieves the highest mLPIPS metric on the Nerfies-HyperNeRF dataset. Shown in the 3$^{rd}$ column, HyperNeRF produces visually implausible results with ghosting effects. Thus we explored incorporating additional regularizations from recent advances in neural rendering. Concretely, we consider the following: (+B)

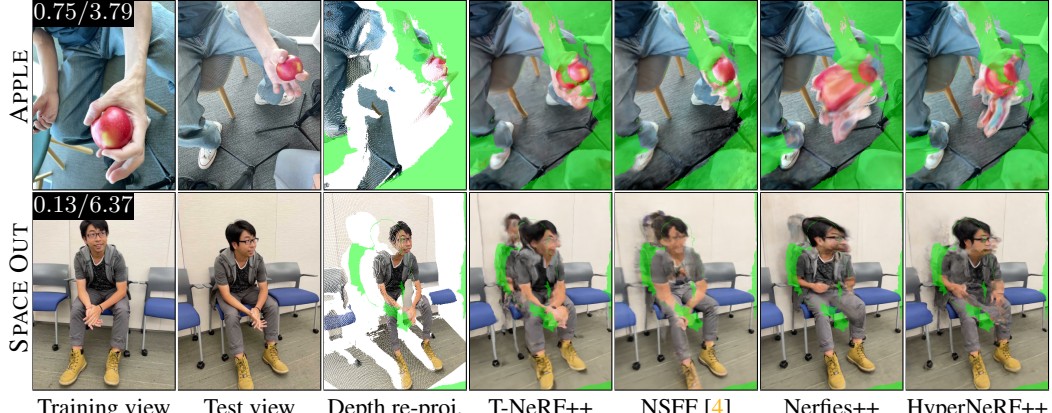

Figure 11: **Qualitative results on the proposed iPhone dataset.** $\Omega/\omega$ metrics of the input sequence are shown on the top-left. The models shown here are trained with all the additional regularizations (+B+D+S) except NSFF. However, existing approaches still struggle to produce high-quality results.

random background compositing [32]; (+D) a depth loss on the ray matching distance [1, 4]; and (+S) a sparsity regularization for scene surface [49]. In Figure 10 (Top), we show quantitative results from the ablation. In Figure 10 (Bottom), we show visualizations of the impact of each regularization. Adding additional regularizations consistently boosts model performance. While we find the random background compositing regularizations particularly helpful, extra depth supervision and surface regularization further improve the quality, *e.g.*, the fan region of the paper windmill.

**Benchmarked results.** In Figure 11, we show qualitative results from our benchmark using the best model settings from the ablation study, denoted as "++". Note that it is non-trivial to apply the same enhancements to NSFF for its NDC formulation so we keep it as-is. We visualize the lidar depth re-projection from the training view (1st column) to the test view (2nd column), as a reference for qualitative comparison (3rd column). Note that the white region is occluded from the input view, whereas the green region is occluded from the most of input video frames. We observe that existing approaches do not handle complex deformation well. For example, all models fail at fusing a valid human shape on the SPACE OUT sequence. In Table 3, we find a similar trend as in the Nerfies-HyperNeRF dataset: the baseline T-NeRF performs the best in terms of mPSNR and mSSIM while HyperNeRF produces the most photorealistic renderings in terms of mLPIPS. The overall synthesis quality and correspondence accuracy of all methods drop considerably compared to the results on the Nerfies-HyperNeRF dataset. Taking Nerfies as an example, it drops $4.4\,\mathrm{dB}$ in mPSNR, $69.6\%$ in mLPIPS, and $40.1\%$ in PCK-T. Our study suggests an opportunity for large improvement when modeling complex motion.

## 6 Discussion and recommendation for future works

In this work, we expose issues in the common practice and establish systematic means to calibrate performance metrics of existing and future works, in the spirit of papers like [50, 51, 52]. We provide initial attempts toward characterizing the difficulty of a monocular video for dynamic view synthesis (DVS) in terms of effective multi-view factors (EMFs). In practice, there are other challenging factors such as variable appearance, lighting condition, motion complexity and more. We leave their characterization for future works. We recommend future works to visualize the input sequences and report EMFs when demonstrating the results. We also recommend future works to evaluate the correspondence accuracy and strive for establishing better correspondences for DVS.

**Acknowledgements.** We would like to thank Zhengqi Li and Keunhong Park for valuable feedback and discussions; Matthew Tancik and Ethan Weber for proofreading. We are also grateful to our pets: Sriracha, Haru, and Mochi, for being good during capture. This project is generously supported in part by the CONIX Research Center, sponsored by DARPA, as well as the BDD and BAIR sponsors.

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
