# Appendix to
# Monocular Dynamic View Synthesis: A Reality Check

**Hang Gao**[1,†]  **Ruilong Li**[1]  **Shubham Tulsiani**[2]  **Bryan Russell**[3]  **Angjoo Kanazawa**[1]

[1]UC Berkeley      [2]Carnegie Mellon University      [3]Adobe Research

## A   Outline

In this Appendix, we describe in detail the following:

- Computation for effective multi-view factors (EMFs) in Section B.
- Computation for co-visibility mask and masked image metrics in Section C.
- Summary of existing works and correspondence readout in Section D.
- Summary of the capture setup and data processing for our iPhone dataset in Section E.
- Summary of the implementation details and remain differences in Section F.
- Additional results on the impact of effective multi-view in Section G.
- Additional results on per-sequence performance breakdown in Section H.
- Additional results on novel-view synthesis in Section I.
- Additional results on inferred correspondence in Section J.

For better demonstration, we strongly recommend visiting our project page for videos of the capture and result visualizations.

## B   Computation for effective multi-view factors (EMFs)

To quantify the amount of effective multi-view in a sequence by the camera and scene motion magnitude, we propose two metrics as effective multi-view factors (EMFs), *i.e.*, the Full EMF $\Omega$ and the angular EMF $\omega$. Note that we design our metrics to be scale-agnostic such that we can compare them across different sequences of different world scales.

As in the main paper, we define a point $\mathbf{x}_t \in \mathbb{S}_t^2$ on the visible object's surface and a camera parameterized by its origin $\mathbf{o}_t$ at time $t \in \mathcal{T}$, where $\mathcal{T}$ is the set of possible time steps.

### B.1   Full EMF $\Omega$: Ratio of camera-scene motion magnitude

We are interested in the relative scale of the camera motion compared to the object. Recall that we define $\Omega$ as the expected ratio over all visible pixels over time,

$$\Omega = \mathop{\mathbb{E}}_{t,t+1 \in \mathcal{T}} \left[ \mathop{\mathbb{E}}_{\mathbf{x}_t \in \mathbb{S}_t^2} \left[ \frac{\|\mathbf{o}_{t+1} - \mathbf{o}_t\|}{\|\mathbf{x}_{t+1} - \mathbf{x}_t\|} \right] \right]. \tag{1}$$

The numerator is trivially computable given the camera information. We thus focus on the denominator, *i.e.*, the foreground 3D scene flow $\mathbf{x}_{t+1} - \mathbf{x}_t$.

We estimate 3D scene flow by combining the known cameras, dense 2D optical flow, and per-frame depth maps. We estimate the 2D optical flow using RAFT [1]. When metric depth is not available, *e.g.*, on previous datasets [2, 3, 4], we use DPT [5] for monocular depth estimation. Additionally, we need a foreground mask for the object, which we obtain through a video segmentation network [6]. For each pixel location $\mathbf{u}_t$ at time $t$ in the foreground mask, we can compute its 3D position $\mathbf{x}_t$ by back-projection with the depth $z_t$. We then get the 2D pixel correspondence at time $t + 1$ by simply

36th Conference on Neural Information Processing Systems (NeurIPS 2022).

following the 2D optical flow $\mathbf{u}_{t+1} = \mathbf{u}_t + \mathbf{f}_{t\to t+1}(\mathbf{u}_t)$, where $\mathbf{f}_{t\to t+1}$ is a bilinearly interpolated forward flow map. After back-projection, we obtain the corresponding 3D point position $\mathbf{x}_{t+1}$ at frame $t+1$. In practice, extra care is needed for handling the unknown depth scale from model prediction and occlusion, discussed next.

**Aligning depth maps of unknown scales.** The DPT [5] model predicts a disparity map in Euclidean space with an unknown scale $a$ and shift $b$. To resolve the scale and shift ambiguity in the predicted disparity maps, we make use of the sparse 3D points extracted by COLMAP. For a frame at time $t$, we first calculate the actual disparity $1/\tilde{z}_t$ of the sparse 3D points by projecting them onto the image, which usually results in sub-pixels. We then bilinearly interpolate the predicted disparity map to get the predicted disparity $1/\tilde{z}_t^*$. Scale $a$ and shift $b$ can then be estimated through linear regression via the relation:

$$\frac{1}{\tilde{z}_t} = a \cdot \frac{1}{\tilde{z}_t^*} + b.$$

Note that the sparse 3D points from COLMAP [7] are all located on the static background. When projecting the sparse 3D points onto the image, some points might be occluded by the moving objects in the foreground. We handle occluded points by fitting $a$ and $b$ using RANSAC [8], which ignores outliers and is robust in practice.

**Handling occlusions.** We identify occlusions using a forward-backward consistency check following the method of Brox *et al.* [9]. We briefly summarize their method here.

Concretely, we identify an occlusion by chaining the forward flow $\mathbf{f}_{t\to t+1}$ and backward flow $\mathbf{f}_{t+1\to t}$ and thresholding based on warp consistency. For those pixels that have inconsistent forward and backward optical flows, defined by regions where chained forward and backward flows result in non-zero flow values, satisfying the following inequality:

$$\|\mathbf{f}_{t\to t+1}(\mathbf{u}_t) + \mathbf{f}_{t+1\to t}(\mathbf{u}_t + \mathbf{f}_{t\to t+1}(\mathbf{u}_t))\|_2^2$$
$$\geqslant 0.01 \cdot (\|\mathbf{f}_{t\to t+1}(\mathbf{u}_t)\|_2^2 + \|\mathbf{f}_{t+1\to t}(\mathbf{u}_t + \mathbf{f}_{t\to t+1}(\mathbf{u}_t))\|_2^2) + 0.5. \tag{2}$$

The occluded pixels, along with the background pixels not belonging to the foreground mask, are excluded from the 3D scene flow computation.

**Discussion.** In practice, we find that the $\Omega$ metric relies on the model estimation quality, in particular, the monocular depth prediction. We therefore design a second metric by measuring camera angular speed $\omega$. With some practical assumptions, it circumvents $\Omega$'s limitation and does not rely on any external model estimates.

## B.2 Angular EMF $\omega$: Camera angular velocity

We propose to measure camera angular speed $\omega$ given the camera parameters, frame rate $N$ and a single 3D look-at point $\mathbf{a}$ obtained by triangulating all cameras, following Nerfies [4]. Recall that $\omega$ is computed as the scaled expectation,

$$\omega = \mathop{\mathbb{E}}_{t,t+1\in\mathcal{T}}\left[\arccos\left(\frac{\langle \mathbf{a}-\mathbf{o}_t, \mathbf{a}-\mathbf{o}_{t+1}\rangle}{\|\mathbf{a}-\mathbf{o}_t\| \cdot \|\mathbf{a}-\mathbf{o}_{t+1}\|}\right)\right] \cdot N. \tag{3}$$

When computing this metric, we assume that (1) the object moves at roughly constant speed, (2) the camera always fixates on the object, and (3) the distance between the camera and the object remains approximately the same over time. All sequences from existing works as well as ours meet these assumptions, except those from the NSFF [2] and NV-DYN [11] datasets. In their case, the cameras are always facing forward, breaking the assumption (2). However, we find that even though cameras are not fixated on the object since they are static, we can still compute the look-at point $\mathbf{a}$ by considering the center of mass of the foreground visible surfaces in 3D. Both datasets provide accurate foreground segmentations and MVS depth, which we use to identify and back-project foreground pixels into 3D space. The final look-at point is computed as the average foreground points over all frames.

Note that existing works only provide extracted frames from each sequence without specifying the frame rate. We identify the frame rates by re-assembling the original video using different FPS candidates and hand-picking the one that results in the most natural object and camera motion, which are verified by the original authors [2, 3, 4]. We document per-sequence FPS for future reference in Table 1.

| | Sequence | #Frames | FPS | $\Omega$ | $\omega$ |
|---|---|---|---|---|---|
| D-NeRF [10] | BOUNCINGBALLS | 150 | 30 | 15.52 | 1945.52 |
| | HELLWARRIOR | 100 | 30 | 10.32 | 2984.15 |
| | HOOK | 100 | 30 | 25.82 | 1996.95 |
| | JUMPINGJACKS | 200 | 30 | 10.64 | 1969.47 |
| | LEGO | 50 | 30 | 17.00 | 2133.78 |
| | MUTANT | 150 | 30 | 12.64 | 1908.67 |
| | STANDUP | 150 | 30 | 14.22 | 2011.31 |
| | TREX | 200 | 30 | 13.70 | 2133.78 |
| HyperNeRF [3] | 3D PRINTER | 207 | 15 | 3.13 | 251.37 |
| | CHICKEN | 164 | 15 | 7.38 | 212.58 |
| | PEEL BANANA | 513 | 15 | 1.26 | 237.66 |
| Nerfies [4] | BROOM | 197 | 15 | 3.40 | 128.54 |
| | CURLS | 57 | 5 | 1.20 | 138.55 |
| | TAIL | 238 | 15 | 3.30 | 160.55 |
| | TOBY SIT | 308 | 15 | 2.18 | 110.51 |
| NSFF [2] | BALLOON1 | 24 | 15 | 2.44 | 57.63 |
| | BALLOON2 | 24 | 30 | 0.76 | 76.97 |
| | DYNAMIC FACE | 24 | 15 | 4.57 | 83.17 |
| | JUMPING | 24 | 30 | 0.68 | 53.79 |
| | PLAYGROUND | 24 | 30 | 0.36 | 71.56 |
| | SKATING | 24 | 30 | 0.79 | 42.76 |
| | TRUCK | 24 | 30 | 0.22 | 19.41 |
| | UMBRELLA | 24 | 15 | 0.62 | 20.66 |
| iPhone (Ours) | APPLE | 475 | 30 | 0.75 | 3.79 |
| | BACKPACK | 180 | 30 | 0.26 | 5.59 |
| | BLOCK | 350 | 30 | 0.04 | 11.50 |
| | CREEPER | 210 | 30 | 0.23 | 14.05 |
| | HANDWAVY | 303 | 30 | 0.05 | 13.66 |
| | HARU | 200 | 60 | 0.30 | 30.32 |
| | MOCHI | 180 | 60 | 0.07 | 14.49 |
| | PAPER WINDMILL | 277 | 30 | 0.38 | 10.71 |
| | PILLOW | 330 | 30 | 0.06 | 13.19 |
| | SPACE OUT | 429 | 30 | 0.13 | 6.37 |
| | SPIN | 426 | 30 | 0.15 | 7.86 |
| | SRIRACHA | 220 | 30 | 0.18 | 18.56 |
| | TEDDY | 350 | 30 | 0.20 | 7.62 |
| | WHEEL | 250 | 30 | 0.03 | 58.45 |

Table 1: **Per-sequence breakdowns of the statistics of different datasets.** As in the main paper, we consider the multi-camera captures from three representative existing datasets: D-NeRF [10], Nerfies [4] and HyperNeRF [3]. We also provide per-sequence breakdowns for both the multi-camera and single-camera captures from our proposed iPhone dataset.

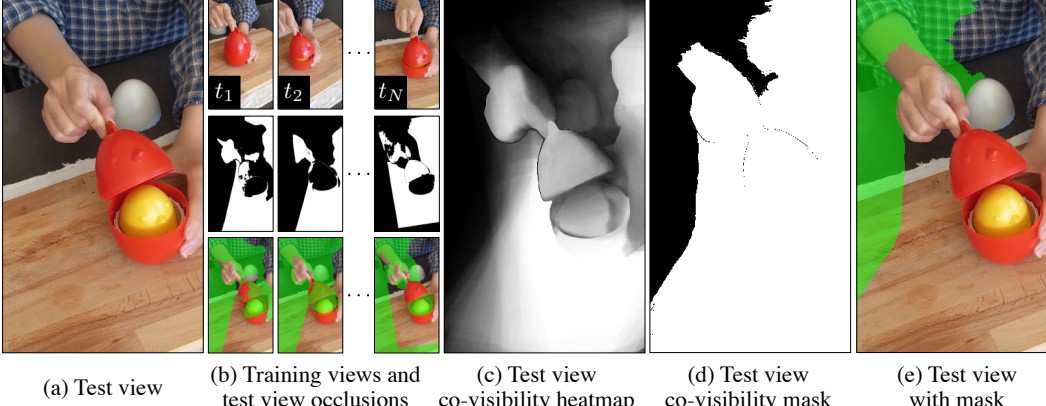

|  (a) Test view | (b) Training views and test view occlusions | (c) Test view co-visibility heatmap | (d) Test view co-visibility mask | (e) Test view with mask |

Figure 1: **Illustration of the computation process for co-visibility.** Given a (a) test view, we first compute its pairwise (b) occlusions in all training views the forward-backward consistency check [9] based on the optical flow estimation from pre-trained RAFT [1]. The occlusions are visualized as binary masks in (b)'s second row, where black color indicates pixels without correspondence. We also visualize their overlays over the original test image. Then by summing up all occlusion maps, we compute the (c) test view co-visibility heatmap, which stores the number of times each test pixel is seen in training frames. Finally, we apply a threshold on the heatmap and obtain a binary (d) co-visibility mask. We also visualize its (e) overlay on the test image. Note that the occlusion maps are usually inaccurate due to noise in optical flow prediction, *e.g.*, they miss the cover of the chicken toy in this example. Our conservative threshold strategy overcomes the noise and ensures that adequately seen regions are included in the final mask.

## C  Computation for co-visibility mask and masked image metrics

Code for both the co-visibility mask and masked image metrics are made publicly available on our project page. In this section, we provide further details for their computation processes.

### C.1  Co-visibility mask

In dynamic scenes, particularly for monocular capture with multi-camera validation, the test view contains regions that may not have been observed at all by the training camera. To circumvent this issue without resorting to camera teleportation, for each pixel in the test image, we propose "co-visibility" masking, which tests how many times a test pixel has been observed in the training images.

We visualize the computation process of the co-visibility mask in Figure 1. Concretely, for each (a) test frame, we first check its (b) occlusion in each training frame by the forward-backward flow consistency check according to Equation 2. We use RAFT [1] for optical flow estimation between each test frame and each training frame. Note that we visualize occlusion as both a binary mask and its overlay on the test image. For occlusion mask visualization, the black color indicates pixels with no correspondence in the training views. We then compute the (c) co-visibility heatmap by simply summing up all test view occlusion masks. This co-visibility heatmap stores the number of times that each pixel is seen in training views. For visualization purpose, we normalize the heatmap by the number of the training frames $N$. Finally, we apply a threshold $\beta$ to the heatmap and obtain a (d) binary co-visibility mask, which we also visualize with (e) its overlay on the test image. We adopt a conservative strategy and set $\beta = \max(5, 0.1 \cdot N)$, meaning that we deem a pixel "seen" during training and valid for evaluation when it is seen in 5 or $10\%$ of training frames, whichever is larger. This strategy ensures high recall in the masking result, *i.e.*, the final co-visible regions are adequately seen during training when the flow estimation is noisy. For example, as shown in the third row of Figure 1 (b), the test view occlusions are inaccurate and miss the red cover of the chicken toy when it is visible in both frames. However, since the red cover is adequately seen over the whole sequence, it is still included in the final co-visibility mask.

## C.2 Masked image metrics

In this work, we propose to only evaluate on regions that are adequately seen during training by co-visibility masking. We employ three masked image metrics, namely mPSNR, mSSIM and mLPIPS, which extend from their original definition, which we discuss next.

**PSNR → mPSNR.** PSNR is originally defined as per-pixel mean squared error (MSE) in the log scale (with a constant negative multiplier). We compute mPSNR by simply taking the average of per-pixel PSNR scores over the masked region.

**SSIM [12] → mSSIM.** Comparing to PSNR, SSIM is defined on the patch level: it considers the structural similarity within each patch. In practice, it is usually implemented as convolutions where kernels are defined by the pixels in each patch. We take inspiration from Liu *et al.* [13] and follow exactly their partial convolution implementation for this operation, where only the masked pixels are accounted for the final result.

**LPIPS [14] → mLPIPS [2, 15, 16].** LPIPS is also defined on patch level. Given two images, it computes their similarity distance in the feature space across different spatial resolution using a pretrained AlexNet model [17]. The final similarity score is the average over all distance maps. To compute mLPIPS, we follow the previous works [2, 15, 16] and first apply the co-visibility mask on the input images by zeroing out the unseen regions. Given the output distance maps at each spatial resolution, we then apply the same mask with downsampling and compute the masked average distance score. It should be noted that the pretrained AlexNet has a receptive field of $195^2$. Thus when the co-visibility mask is small (most of the pixels are not seen during training), this metric can be artificially low due to the zeroing operation.

## D Correspondence readout from existing works

In this section, we first review the formulation of the existing works and then describe the computation to read out correspondence from these models.

### D.1 Formulation of existing works

A neural radiance field (NeRF) [18] represents a *static* scene as a continuous volumetric field $F$ that transforms a point's position $\mathbf{x}$ and auxiliary variables $\mathbf{w}$ (*e.g.*, view direction, latent appearance vector) to color $\mathbf{c}$ and density $\sigma$,

$$F : (\mathbf{x}, \mathbf{w}) \mapsto (\mathbf{c}, \sigma). \tag{4}$$

Here we briefly review representative approaches that extend NeRFs to dynamic scenes.

**Nerfies [4] and HyperNeRF [3].** Similarly to traditional non-rigid reconstruction methods that explains non-rigid scenes with a static *canonical* space and a per-frame deformation model [19], Nerfies [4] capture a non-rigid scene with one canonical NeRF $F$ and a per-time step view-to-canonical deformation $W_{t\to c}$ that takes a point $\mathbf{x}$ with a time-conditioned latent vector $\boldsymbol{\varphi}_t$ to a canonical point $\mathbf{x}_c$,

$$W_{t\to c} : (\mathbf{x}, \boldsymbol{\varphi}_t) \mapsto \mathbf{x}_c. \tag{5}$$

At each time step the resulting volumetric field is $F_t = F \circ W_{t\to c}$. HyperNeRF [3] addresses topological change on top of Nerfies by outputting a two-dimensional "ambient" coordinate $\mathbf{w}$ encoding the topological change in addition to the canonical point $\mathbf{x}_c$,

$$W_{t\to c} : (\mathbf{x}, \boldsymbol{\varphi}_t) \mapsto (\mathbf{x}_c, \mathbf{w}). \tag{6}$$

These two output variables are passed to the (topologically varying) canonical space mapping $F$.

**Time-conditioned NeRF and NSFF [2].** Another way to handle non-rigid scenes is to directly map space-time to the output color and density by a time-conditioned latent vector $\boldsymbol{\varphi}_t$, which we refer to as T-NeRF:

$$F_t : (\mathbf{x}, \boldsymbol{\varphi}_t) \mapsto (\mathbf{c}, \sigma). \tag{7}$$

Note that since T-NeRF implicitly handles deformation, it is difficult to compute correspondences over time. NSFF [2] augments T-NeRF's implicit function $F_t$ to output an explicit scene flow field $W_{t \to t+\delta}$ between adjacent time steps $t$ and $t + \delta$,

$$W_{t \to t+\delta} : \mathbf{x} \mapsto \mathbf{x}', \ \ \delta \in \{+1, -1\}. \tag{8}$$

This explicit flow field is used to regularize motion and, as shown below, can be chained to compute long-range point correspondences across views and times.

### D.2 Correspondence readout

Our goal is to find view-to-view correspondences such that given a set of key-points on a source image at time $t_1$, we can find their correspondence on a target image at time $t_2$.

For clarity, we start with assuming a known 3D view-to-view warp $W_{t_1 \to t_2}$, outlined in the last sub-section. The 2D correspondence $\mathbf{u}_{t_2}$ given $\mathbf{u}_{t_1}$ can be obtained by three steps, which we describe as "warp-integrate-project". In the "warp" step, given the pixel location $\mathbf{u}_{t_1}$ and camera $\pi_{t_1}$, we sample points on the ray passing from the camera center through the pixel $\pi_{t_1}^{-1}(\mathbf{u}_{t_1})$. Then, we warp the sampled points toward their 3D correspondences in the target frame using the known 3D warp $W_{t_1 \to t_2}$. In the "integrate" step, we compute the expected 3D location for the source samples weighted by the probability mass $w_{t_1}$ by volume rendering, as per NeRF [18]. We can use densities from either source or target frame, a choice that we find insensitive in practice. In our formulation, we use the densities from the source frame. Finally, in the "project" step, we project the expected 3D location to the target frame through the target camera $\pi_{t_2}$. The "warp-integrate-project" process can be written as

$$\mathbf{u}_{t_2} = \pi_{t_2} \left( \mathop{\mathbb{E}}_{\mathbf{x}_{t_1} \in \pi_{t_1}^{-1}(\mathbf{u}_{t_1})} \left[ w_{t_1}(\mathbf{x}_{t_1}) \cdot W_{t_1 \to t_2}(\mathbf{x}_{t_1}) \right] \right). \tag{9}$$

Note that there are also other alternatives such as "warp-project-integrate" where integration happens after projecting warped points to 2D. We find in practice that these different approaches make little difference to the final results when the surface is dense such that $w_t$ is concentrated near one point (almost one-hot) for each ray.

**Nerfies [4] and HyperNeRF [3].** We can compose $W_{t_1 \to t_2}$ by an inverse map $W_{t_1 \to c}$ and a forward map $\tilde{W}_{c \to t_2}$,

$$W_{t_1 \to t_2}(\mathbf{x}_{t_1}) = \tilde{W}_{c \to t_2}(W_{t_1 \to c}(\mathbf{x}_{t_1})). \tag{10}$$

We solve for the forward map given the inverse map through optimization:

$$\tilde{W}_{c \to t}(\mathbf{x}_t) = \arg\min_{\mathbf{x}_c} \| W_{t \to c}(\mathbf{x}_t) - \mathbf{x}_c \|_2^2. \tag{11}$$

We use the Broyden solver for root-finding, as per SNARF [20], and initialize $\mathbf{x}_c$ with $\mathbf{x}_t$.

**NSFF [2].** We can compose $W_{t_1 \to t_2}$ by chaining the scene flow predictions through time. Concretely we have

$$W_{t_1 \to t_2}(\mathbf{x}_{t_1}) = W_{t_2 - 1 \to t_2} \left( \cdots W_{t_1 + 1 \to t_1 + 2} \big( W_{t_1 \to t_1 + 1}(\mathbf{x}_{t_1}) \big) \right). \tag{12}$$

## E Summary of the capture setup and data processing for our iPhone dataset

Our capture setup has 7 multi-camera captures (MV) and 7 single camera captues (SV). We evaluate novel-view synthesis on the multi-camera captures and correspondence on all captures.

**Multi-camera captures.** For multi-camera captures, we employ three cameras: one hand-held camera to capture monocular video for training and two stationary mounted cameras for validation. The two validation cameras face inward from two distinct viewpoints with large baseline. This wide-baseline setup enables us to better evaluate the shape modeling quality for novel-view synthesis. We use the "Record3D" app [21] on iPhone to record both RGB and depth information at each time step. Note that we only collect depth information for training views given that we will only use the

depths for supervision. We discuss the preprocessing procedure for the training video sequence in the "Single-camera captures" paragraph below.

To synchronize multiple cameras, we leverage the "audio-based multi-camera synchronization" functionality in Adobe Premiere Pro, as per [22], which achieves millisecond-level accuracy. In Figure 2, we show visualizations of our multi-camera captures after time synchronization. To ensure that our input sequence covers most of the scene regions in evaluation, we intentionally move the training camera in front of each test camera at certain frames. When we do so, that particular test frame is excluded due to severe occlusion (shown as "Excluded" in the figure). For the WHEEL sequence (last row), we only employ the right camera due to the limited physical space to set up the multi-camera rig in that scene.

After time synchronization, we calibrate the multi-camera system. The Record3D app provides camera parameters and poses at each time step, but the poses only relate to each other *within* each capture sequence. In fact, each camera pose is recorded as relative pose to the first frame in each sequence, with the first pose being identity. We therefore need to solve the relative $SE(3)$ transforms between the first frame in each test sequence with respect to the first training frame. This problem can be formulated as a Perspective-n-Point (PnP) problem where, given a set of 3D points and their corresponding 2D pixels in two sequences, we aim to solve the camera pose. In practice, given a training RGBD frame and a testing RGB frame, we compute a set of 2D correspondences by SIFT feature matching [23] and obtain their 3D point positions (in the training sequence's world space) by back-projecting the 2D keypoints with the training frame depth map. This process is repeated for all time steps. We exploit our problem structure by constraining the camera poses within each test sequence to be the same, *i.e.*, static camera. We use the RANSAC PnP solver [24] in OpenCV [25].

**Single-camera captures.** We treat the single-camera captures as the training sequence in our multi-camera capture setup. In effect, the single-camera capture setup will not have validation data for novel-view synthesis evaluation. We preprocess the depth data for the training sequence by applying a Sobel filter [26] to filter out inaccurate depth values around object edges. In Figure 3, we visualize our depth data before and after filtering. We find that NeRF is particularly sensitive to depth noise and this filtering step is necessary. Finally, we manually annotate keypoints for correspondence evaluation. For sequences of humans and quadrupeds (dogs or cats), we annotate keypoints based on the skeleton defined in the COCO challenge [27] and StanfordExtra [28]. For sequences that focus on more general objects (*e.g.*, our BLOCK and TEDDY sequences), we manually identify and annotate 5 to 15 trackable keypoints across frames. We visualize keypoint annotations (with skeleton if available) for both our proposed iPhone dataset and the Nerfies-HyperNeRF dataset in Figure 4.

Note that both Nerfies [4] and HyperNeRF [3] use background regularization which requires a point cloud of the background static scene. We first extract the object mask over time by MTTR, an off-the-shelf video segmentation network [6], which takes a text prompt of the foreground object as input. Since our foreground objects are quite diverse (*e.g.*, backpack and block), the segmentation results are usually noisy. Thus we apply TSDF Fusion [29] to the background point clouds over the whole sequence to get a completed background point cloud. We find that this point cloud can be noisy when segmentation fails, and that it is necessary to manually filter the background point cloud to make sure that it does not include any foreground regions. We consider this manual process a weakness of the previous background regularization [4].

# F   Summary of the implementation details and remaining differences

To ensure a fair comparison, we align numerous training details between the models that we investigate in this paper: T-NeRF, NSFF [2], Nerfies [4] and HyperNeRF [3]. Code and checkpoints are available on our project page.

To start with, we align the total number of rays seen during training. We add support of ray undistortion [3] in the third-party implementation of NSFF [30] to make sure that the training rays are the same across codebases. All models are trained with view-dependency modeling turned on. We did not find appearance encoding [31] helpful in terms of quantitative results. This might due to the lighting difference between training and validation captures – a common issue in evaluation discussed in mip-NeRF 360 [32].

| Method | BROOM $\Omega = 3.40, \omega = 128.54$ | | | CURLS $\Omega = 1.20, \omega = 138.55$ | | | TAIL $\Omega = 3.30, \omega = 160.55$ | | | TOBY SIT $\Omega = 2.18, \omega = 110.51$ | | |
|---|---|---|---|---|---|---|---|---|---|---|---|---|
| | PSNR↑ | SSIM↑ | LPIPS↓ | PSNR↑ | SSIM↑ | LPIPS↓ | PSNR↑ | SSIM↑ | LPIPS↓ | PSNR↑ | SSIM↑ | LPIPS↓ |
| Nerfies [4] | 19.40 | - | 0.323 | - | - | - | - | - | - | - | - | - |
| Nerfies (repo) | 19.40 | - | 0.325 | **24.40** | - | 0.392 | **21.90** | - | 0.245 | 18.44 | - | 0.384 |
| Nerfies (our reimpl.) | **19.70** | 0.216 | 0.296 | 24.04 | 0.670 | 0.245 | 21.79 | 0.314 | 0.236 | 18.48 | 0.355 | 0.375 |
| HyperNeRF [3] | 19.30 | - | **0.296** | - | - | - | - | - | - | - | - | - |
| HyperNeRF (repo) | 19.30 | - | 0.308 | **24.60** | - | 0.363 | 22.10 | - | **0.226** | 18.40 | - | **0.330** |
| HyperNeRF (our reimpl.) | **19.36** | 0.210 | 0.314 | 24.59 | 0.686 | 0.247 | 22.16 | 0.329 | 0.231 | 18.41 | 0.345 | 0.339 |

| Method | 3D PRINTER $\Omega = 3.13, \omega = 251.37$ | | | CHICKEN $\Omega = 7.38, \omega = 212.58$ | | | PEEL BANANA $\Omega = 1.26, \omega = 237.66$ | | |
|---|---|---|---|---|---|---|---|---|---|
| | PSNR↑ | SSIM↑ | LPIPS↓ | PSNR↑ | SSIM↑ | LPIPS↓ | PSNR↑ | SSIM↑ | LPIPS↓ |
| Nerfies [4] | 20.20 | - | **0.115** | 26.00 | - | 0.084 | 21.70 | - | **0.157** |
| Nerfies (repo) | 20.20 | - | 0.118 | **26.80** | - | 0.081 | **22.00** | - | 0.179 |
| Nerfies (our reimpl.) | **20.30** | 0.639 | 0.115 | 26.54 | 0.823 | 0.079 | 21.11 | 0.693 | 0.174 |
| HyperNeRF [3] | 20.00 | - | 0.111 | 26.90 | - | 0.079 | **23.30** | - | 0.133 |
| HyperNeRF (repo) | 20.10 | - | **0.110** | 27.70 | - | **0.076** | 22.20 | - | 0.140 |
| HyperNeRF (our reimpl.) | **20.12** | 0.638 | 0.110 | 27.74 | 0.834 | 0.077 | 22.25 | 0.729 | 0.144 |

Table 2: **Our re-implementation reproduces Nerfies [4]'s and HyperNeRF [3]'s results.** The official numbers for both Nerfies and HyperNeRF are taken from the HyperNeRF paper. Our results matches closely to their numbers and the ones that we obtained by running the officially released repositories (denoted as "repo"). All models are trained under the teleporting setting.

| Method | JUMPING $\Omega = 0.68, \omega = 53.79$ | | |
|---|---|---|---|
| | PSNR↑ | SSIM↑ | LPIPS↓ |
| NSFF (repo) | 27.41 | 0.900 | 0.057 |
| NSFF (our reimpl.) | **27.80** | **0.908** | **0.051** |

Table 3: **Our modified third-party re-implementation reproduces NSFF [2]'s results on one sequence from Yoon *et al.* [11].** Due to the absence of per-sequence results in the original paper, we compare to the numbers that we obtained by evaluating the officially released checkpoints (denoted as "repo"). Our results matches closely to their numbers. All models are trained under the teleporting setting.

T-NeRF, Nerfies, and HyperNeRF share the exact same training setup since they are implemented within our codebase. We follow the hyper-parameters specified in their official repositories. We use a batch size $B = 6144$ for a total number of iterations $N = 2.5 \times 10^5$, optimized by ADAM [33] with an initial learning rate $\eta = 1 \times 10^{-3}$ exponentially decayed to $1 \times^{-4}$ at the end. We use this training recipe for all of our experiments across all datasets. On 4 NVIDIA RTX A4000 or 2 NVIDIA A100 GPUs with 24GB memory, it takes roughly 12 hours to train a T-NeRF and 24 hours to train a Nerfies or a HyperNeRF. In Table 2, we show that our codebase reproduces the numbers from the original papers and official repositories.

Due to no publicly available code to train NSFF on the Nerfies-HyperNeRF dataset. We adapt and extend the third-party implementation of NSFF (which we find to perform better than the official repo [34]). We confirm the finding from HyperNeRF that the default hyper-parameters in the NSFF paper are not suitable for long video sequences, and use their hyper-parameters instead. In Table 3, we check on one sequence that our modified re-implementation of NSFF can reproduce the numbers from the ones we obtain by running the released code. On 1 NVIDIA RTX A4000 or NVIDIA A100 GPU, it takes roughly 72 to train a NSFF. With better implementation, we hypothesize that the training process can be largely accelerated.

While we try to ensure the fairness in our comparison, there are still four main remaining differences, namely: (1) static scene stablization, (2) sampling and rendering, (3) NeRF coordinates, and (4) flow supervision. First, Nerfies and HyperNeRF use additional background points from SfM system as supervision to stabilize the static region of the scene, which we find sensitive to foreground segmentation errors as mentioned in Section E. On the other hand, NSFF stabilizes the static region by composing the samples from a time-invariant static NeRF and a time-varying dynamic NeRF. Second, Nerfies and HyperNeRF sample $S = 128$ points during the coarse stage, and another $2S$ points

during the fine stage, evaluating $3S = 384$ points in total. NSFF, on the other hand, only samples $S$ points for dynamic NeRF and another $S$ points for static NeRF, without coarse-to-fine sampling, evaluating $2S = 256$ points in total. Third, Nerfies and HyperNeRF sample points in world space, while NSFF samples in normalized device coordinates (NDC), which can cause issues when applying to non-forward-facing scenes like the ones we use in this paper. Finally, NSFF uses additional optical flow supervision, while Nerfies and HyperNeRF do not. In fact, we consider the fact that NSFF can leverage correspondence supervision as a merit in the sense that it is non-trivial to apply optical flow supervision to Nerfies and HyperNeRF since their warp representation is not fully invertible.

## G  Additional results on the impact of effective multi-view

In Figure 5, we provide more qualitative comparisons between models that are trained with and without camera teleportation on the Nerfies-HyperNeRF dataset.

## H  Additional results on per-sequence quantitative performance breakdown

| | BROOM ($\Omega$ =2.57, $\omega$ =60.4) | | | | CURLS ($\Omega$ =0.90, $\omega$ =118.7) | | | |
|---|---|---|---|---|---|---|---|---|
| Method | mPSNR↑ | mSSIM↑ | mLPIPS↓ | PCK-T↑ | mPSNR↑ | mSSIM↑ | mLPIPS↓ | PCK-T↑ |
| T-NeRF | 20.04(20.17) | **0.344**(0.257) | 0.590(0.624) | - | 21.86(21.75) | 0.677(0.597) | 0.284(0.341) | - |
| NSFF | **20.36**(20.46) | 0.335(0.247) | **0.776**(0.813) | 0.119 | 18.74(18.85) | 0.616(0.531) | **0.378**(0.423) | 0.212 |
| Nerfies | 19.34(19.51) | 0.293(0.202) | 0.294(0.327) | 0.460 | **23.28**(23.03) | **0.707**(0.630) | 0.220(0.266) | 0.782 |
| HyperNeRF | 19.04(19.23) | 0.288(0.197) | 0.279(0.313) | **0.471** | 23.13(22.98) | 0.700(0.625) | 0.220(0.266) | **0.838** |

| | TAIL ($\Omega$ =1.31, $\omega$ =28.6) | | | | TOBY-SIT ($\Omega$ =1.28, $\omega$ =26.4) | | | |
|---|---|---|---|---|---|---|---|---|
| Method | mPSNR↑ | mSSIM↑ | mLPIPS↓ | PCK-T↑ | mPSNR↑ | mSSIM↑ | mLPIPS↓ | PCK-T↑ |
| T-NeRF | **22.56**(22.11) | 0.460(0.385) | 0.305(0.365) | - | 18.53(18.53) | 0.428(0.330) | 0.421(0.471) | - |
| NSFF | 21.94(21.72) | **0.461**(0.388) | **0.522**(0.579) | 0.323 | **18.66**(18.65) | **0.429**(0.329) | **0.600**(0.634) | 0.666 |
| Nerfies | 21.46(21.17) | 0.385(0.305) | 0.213(0.261) | **0.645** | 18.45(18.41) | 0.423(0.326) | 0.249(0.307) | **0.914** |
| HyperNeRF | 21.54(21.13) | 0.382(0.301) | 0.218(0.263) | 0.623 | 18.40(18.33) | 0.422(0.324) | 0.242(0.300) | 0.883 |

| | 3DPRINTER ($\Omega$ =1.22, $\omega$ =59.4) | | | | CHICKEN ($\Omega$ =1.52, $\omega$ =33.5) | | | |
|---|---|---|---|---|---|---|---|---|
| Method | mPSNR↑ | mSSIM↑ | mLPIPS↓ | PCK-T↑ | mPSNR↑ | mSSIM↑ | mLPIPS↓ | PCK-T↑ |
| T-NeRF | **19.69**(18.60) | **0.665**(0.591) | 0.205(0.238) | - | **25.54**(24.41) | **0.802**(0.764) | 0.131(0.158) | - |
| NSFF | 16.89(16.26) | 0.526(0.426) | **0.443**(0.492) | 0.797 | 21.47(20.72) | 0.671(0.619) | **0.290**(0.325) | 0.604 |
| Nerfies | 19.67(18.81) | 0.661(0.588) | 0.148(0.175) | **0.998** | 23.78(22.71) | 0.784(0.742) | 0.114(0.142) | 0.978 |
| HyperNeRF | 19.58(18.73) | 0.656(0.583) | 0.147(0.175) | 0.994 | 24.90(23.88) | 0.792(0.753) | 0.101(0.125) | **1.000** |

| | PEEL-BANANA ($\Omega$ =0.33, $\omega$ =33.8) | | | |
|---|---|---|---|---|
| Method | mPSNR↑ | mSSIM↑ | mLPIPS↓ | PCK-T↑ |
| T-NeRF | **22.64**(22.07) | **0.787**(0.721) | 0.142(0.185) | - |
| NSFF | 18.68(18.62) | 0.613(0.530) | **0.293**(0.335) | 0.233 |
| Nerfies | 19.97(19.85) | 0.677(0.609) | 0.161(0.206) | 0.514 |
| HyperNeRF | 21.34(21.08) | 0.707(0.641) | 0.135(0.173) | **0.540** |

Table 4: **Per-scene breakdowns of the quantitative results on the Nerfies-HyperNeRF dataset.** Numbers in gray are calculated without using the co-visibility mask. All models are trained under the non-teleporting setting.

We document the per-sequence quantitative performances of different models on both the Nerfies-HyperNeRF dataset (under non-teleporting setting) in Table 4 and the proposed iPhone dataset in Table 5.

## I  Additional results on novel-view synthesis

We provide additional novel-view synthesis qualitative results under the non-teleporting setting. In Figure 6, we show qualitative results on the Nerfies-HyperNeRF dataset. In Figure 7, we show qualitative results on the multi-camera captures from the proposed iPhone dataset. All models except NSFF [2] are trained with all the additional regularizations that we find helpful through ablation,

| | APPLE ($\Omega$ =0.75, $\omega$ =3.8) | | | | BLOCK ($\Omega$ =0.04, $\omega$ =11.5) | | | |
|---|---|---|---|---|---|---|---|---|
| Method | mPSNR↑ | mSSIM↑ | mLPIPS↓ | PCK-T↑ | mPSNR↑ | mSSIM↑ | mLPIPS↓ | PCK-T↑ |
| T-NeRF | 17.43(15.98) | 0.728(0.375) | **0.508**(0.598) | - | 17.52(17.15) | 0.669(0.521) | 0.346(0.449) | - |
| NSFF | 17.54(16.50) | 0.750(0.432) | 0.478(0.548) | **0.599** | 16.61(16.34) | 0.639(0.494) | 0.389(0.482) | **0.274** |
| Nerfies | **17.64**(16.34) | 0.743(0.411) | 0.478(0.563) | 0.318 | **17.54**(17.35) | **0.670**(0.528) | 0.331(0.424) | 0.216 |
| HyperNeRF | 16.47(16.07) | **0.754**(0.425) | 0.414(0.505) | 0.132 | 14.71(14.93) | 0.606(0.460) | **0.438**(0.517) | 0.180 |

| | PAPER-WINDMILL ($\Omega$ =0.38, $\omega$ =10.7) | | | | SPACE-OUT ($\Omega$ =0.13, $\omega$ =6.4) | | | |
|---|---|---|---|---|---|---|---|---|
| Method | mPSNR↑ | mSSIM↑ | mLPIPS↓ | PCK-T↑ | mPSNR↑ | mSSIM↑ | mLPIPS↓ | PCK-T↑ |
| T-NeRF | **17.55**(17.55) | 0.367(0.349) | 0.258(0.268) | - | 17.71(17.04) | 0.591(0.521) | **0.377**(0.438) | - |
| NSFF | 17.34(17.35) | 0.378(0.362) | 0.211(0.218) | 0.113 | 17.79(17.25) | 0.622(0.560) | 0.303(0.359) | 0.812 |
| Nerfies | 17.38(17.39) | **0.382**(0.366) | 0.209(0.215) | 0.107 | **17.93**(18.10) | 0.605(0.546) | 0.320(0.369) | **0.859** |
| HyperNeRF | 14.94(14.98) | 0.272(0.254) | **0.348**(0.361) | **0.163** | 17.65(17.79) | **0.636**(0.578) | 0.341(0.390) | 0.598 |

| | SPIN ($\Omega$ =0.15, $\omega$ =7.9) | | | | TEDDY ($\Omega$ =0.20, $\omega$ =7.6) | | | |
|---|---|---|---|---|---|---|---|---|
| Method | mPSNR↑ | mSSIM↑ | mLPIPS↓ | PCK-T↑ | mPSNR↑ | mSSIM↑ | mLPIPS↓ | PCK-T↑ |
| T-NeRF | 19.16(18.17) | 0.567(0.441) | **0.443**(0.490) | - | 13.71(13.32) | **0.570**(0.331) | 0.429(0.565) | - |
| NSFF | 18.38(16.97) | **0.585**(0.445) | 0.309(0.380) | **0.177** | 13.65(12.91) | 0.557(0.302) | 0.372(0.508) | **0.801** |
| Nerfies | **19.20**(18.59) | 0.561(0.436) | 0.325(0.377) | 0.115 | **13.97**(13.91) | 0.568(0.327) | 0.350(0.479) | 0.775 |
| HyperNeRF | 17.26(16.52) | 0.540(0.414) | 0.371(0.437) | 0.083 | 12.59(12.78) | 0.537(0.304) | **0.527**(0.635) | 0.291 |

| | WHEEL ($\Omega$ =0.03, $\omega$ =58.5) | | | |
|---|---|---|---|---|
| Method | mPSNR↑ | mSSIM↑ | mLPIPS↓ | PCK-T↑ |
| T-NeRF | **15.65**(14.42) | **0.548**(0.405) | 0.292(0.363) | - |
| NSFF | 13.82(13.19) | 0.458(0.312) | 0.310(0.366) | 0.394 |
| Nerfies | 13.99(13.35) | 0.455(0.307) | 0.310(0.366) | **0.408** |
| HyperNeRF | 14.59(13.31) | 0.511(0.359) | **0.331**(0.402) | 0.346 |

Table 5: **Per-scene breakdowns of the quantitative results on the proposed iPhone dataset.** Numbers in gray are calculated without using the co-visibility mask.

| Method | CREEPER | BACKPACK | HANDWAVY | HARU | MOCHI | PILLOW | SRIRACHA | MEAN |
|---|---|---|---|---|---|---|---|---|
| NSFF [2] | 0.560 | 0.269 | 0.178 | 0.699 | 0.624 | 0.154 | 0.616 | 0.443 |
| Nerfies++ | **0.708** | **0.329** | 0.685 | **0.942** | **0.908** | 0.575 | **0.737** | **0.698** |
| HyperNeRF++ | 0.702 | 0.260 | **0.708** | 0.817 | 0.891 | **0.602** | 0.617 | 0.657 |

Table 6: **Additional quantitative results of the PCK-T evaluation on the single-camera captures from the proposed iPhone dataset.** The correspondence evaluation is applicable when multi-camera validation is not available. All numbers are computed with $\alpha = 0.05$.

denoted with "++" to distinguish with the original models. In Figure 8, we show qualitative results on the single-camera captures from the proposed iPhone dataset. We render novel views using the camera pose from the first captured frame. Finally, in Figure 9, we show the rendering results with and without co-visibility mask applied.

## J    Additional results on inferred correspondence

In Table 6, we provide additional quantitative results of the inferred correspondence on the single-camera captures from the proposed iPhone dataset. In Figure 10 and 11, we provide additional qualitative results of the inferred correspondence on both the Nerfies-iPhone dataset and the proposed iPhone dataset. Note that all models are trained with additional regularizations on the proposed iPhone dataset except NSFF.

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

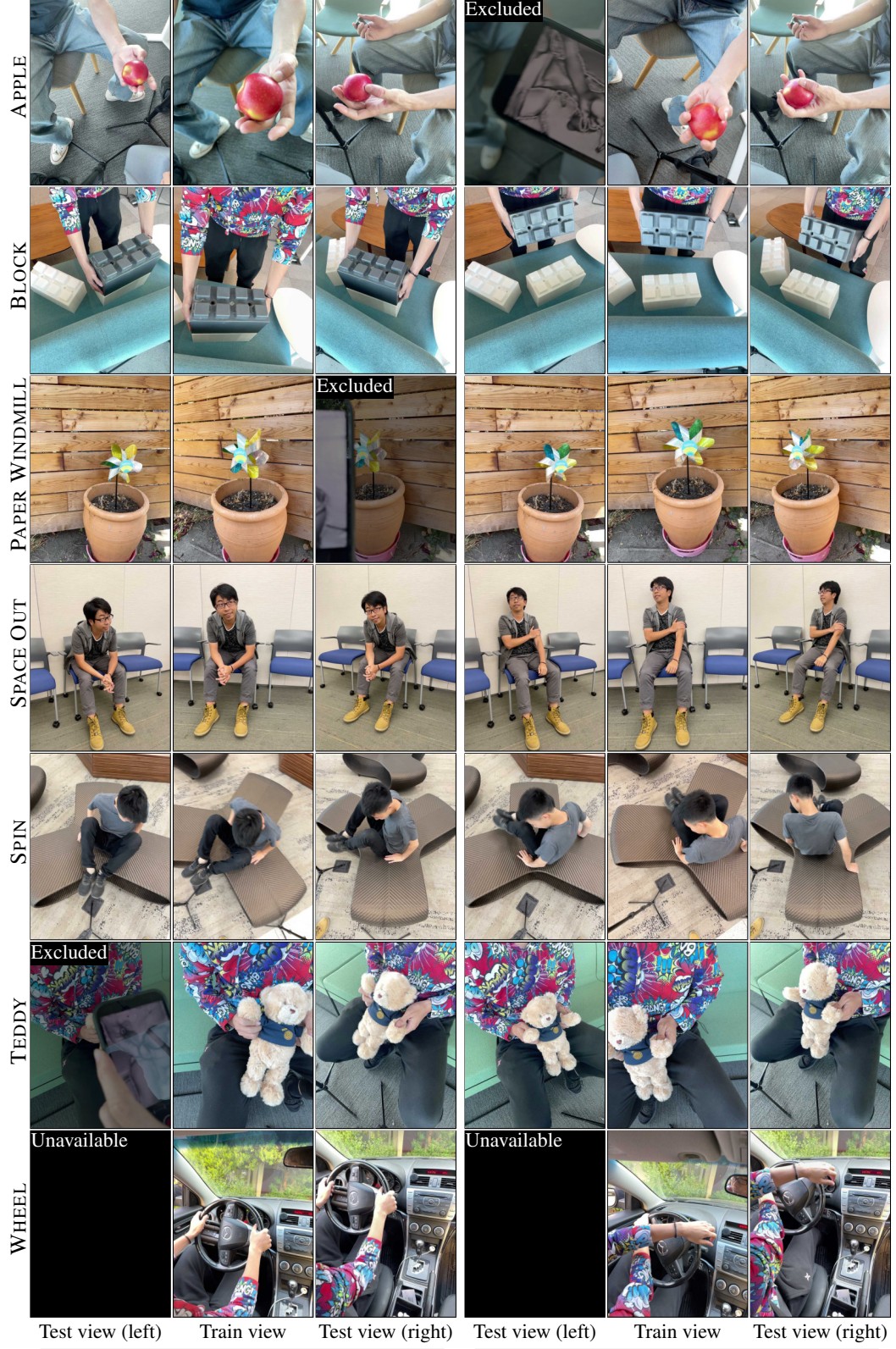

Test view (left)    Train view    Test view (right)    Test view (left)    Train view    Test view (right)

Figure 2: **Visualizations of the multi-camera captures after time synchronization from the proposed iPhone dataset.** In each row, we visualize the frames from both the training camera and two testing cameras at two time steps. We intentionally move the training camera in front of each test camera at certain times to ensure that our input sequence covers most of the scene in evaluation. When a particular test frame depicts the training camera, we exclude the test frame (denoted as "Excluded"). For the WHEEL sequence in the last row, we only employ the right test camera due to limited space to set up the multi-camera rig.

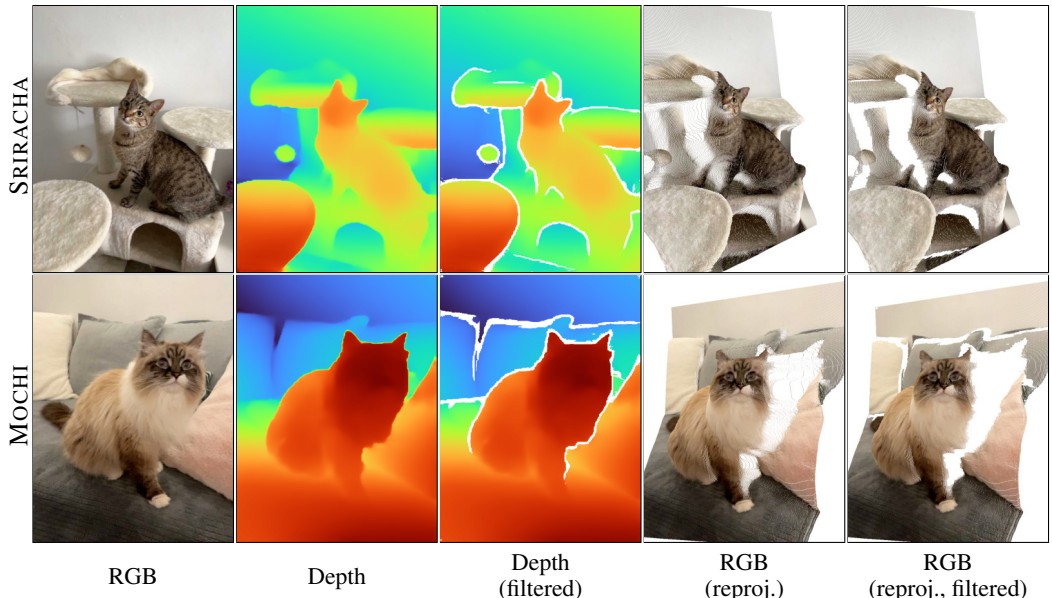

| RGB | Depth | Depth (filtered) | RGB (reproj.) | RGB (reproj., filtered) |

Figure 3: **Visualizations of the depth filtering during data preprocessing of the proposed iPhone dataset.** The depth sensing is particularly noisy around the object edges, which we filtered out by Sobel filter [26]. We visualize the re-projected RGB image with the original (2nd column) or filtered depth (3rd column) from the captured view to the first view in each sequence at the last two columns. Without filtering (4th column), there are erroneous floaters which cause too much noise for training supervision. With filtering (last column), we have a crisper depth map which is used for improving the state-of-the-art methods.

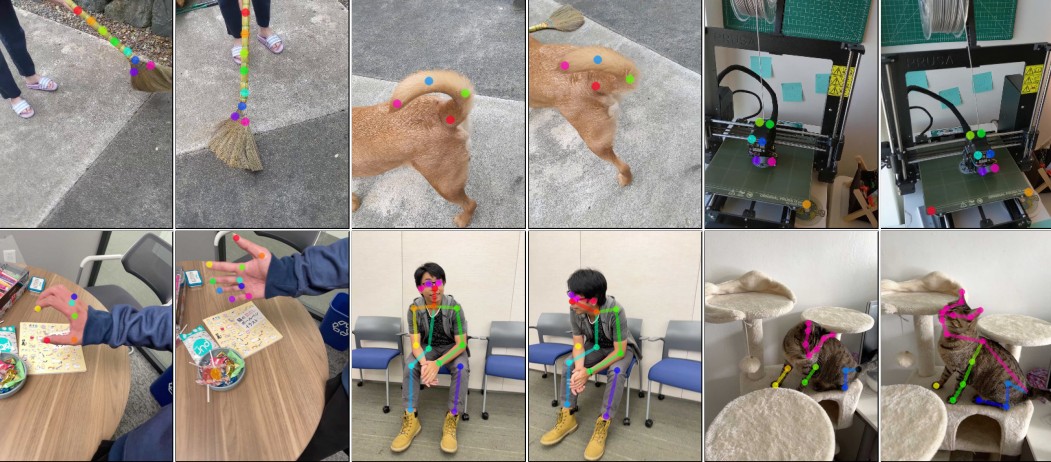

Figure 4: **Visualizations of the keypoint annotation during data preprocessing of the proposed iPhone dataset.** We manually annotate keypoints for correspondence evaluation. For sequences of humans and quadrupeds (dogs or cats), we annotate based on the skeleton defined in the COCO challenge [27] and StanfordExtra [28]. For sequences that focus on more general objects, we manually identify and annotate 5 to 15 trackable keypoints across frames.

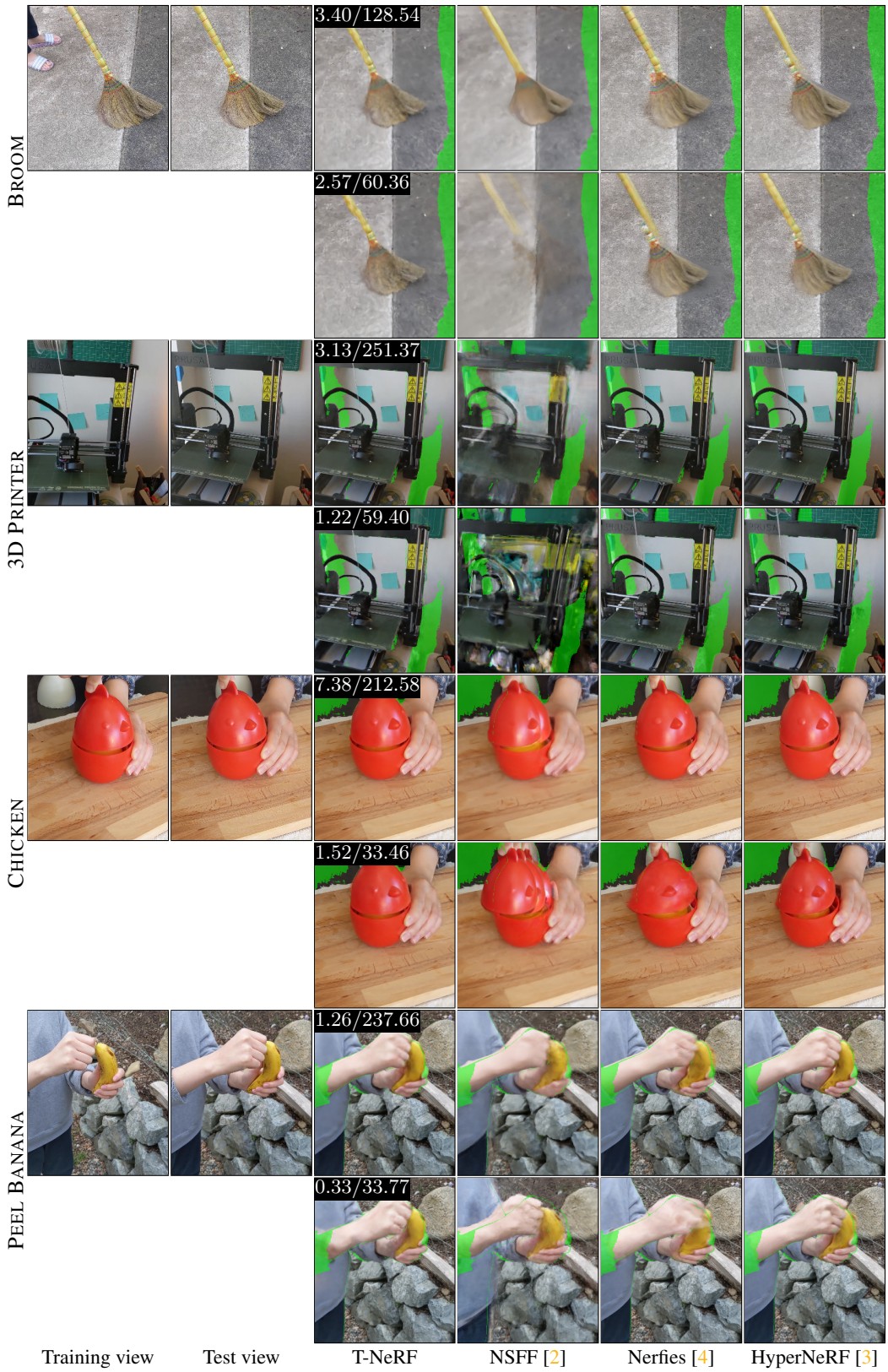

| Training view | Test view | T-NeRF | NSFF [2] | Nerfies [4] | HyperNeRF [3] |

Figure 5: **Additional qualitative results on the impact of effective multi-view on the Nerfies-HyperNeRF dataset.** $\Omega/\omega$ metrics of the input sequence are shown on the top-left. We compare the existing camera teleporting setting and our non-teleporting setting. For every two rows, we show the results trained with and without camera teleportation in the first and second rows. Two settings use the same set of co-visibility masks computed from common training images.

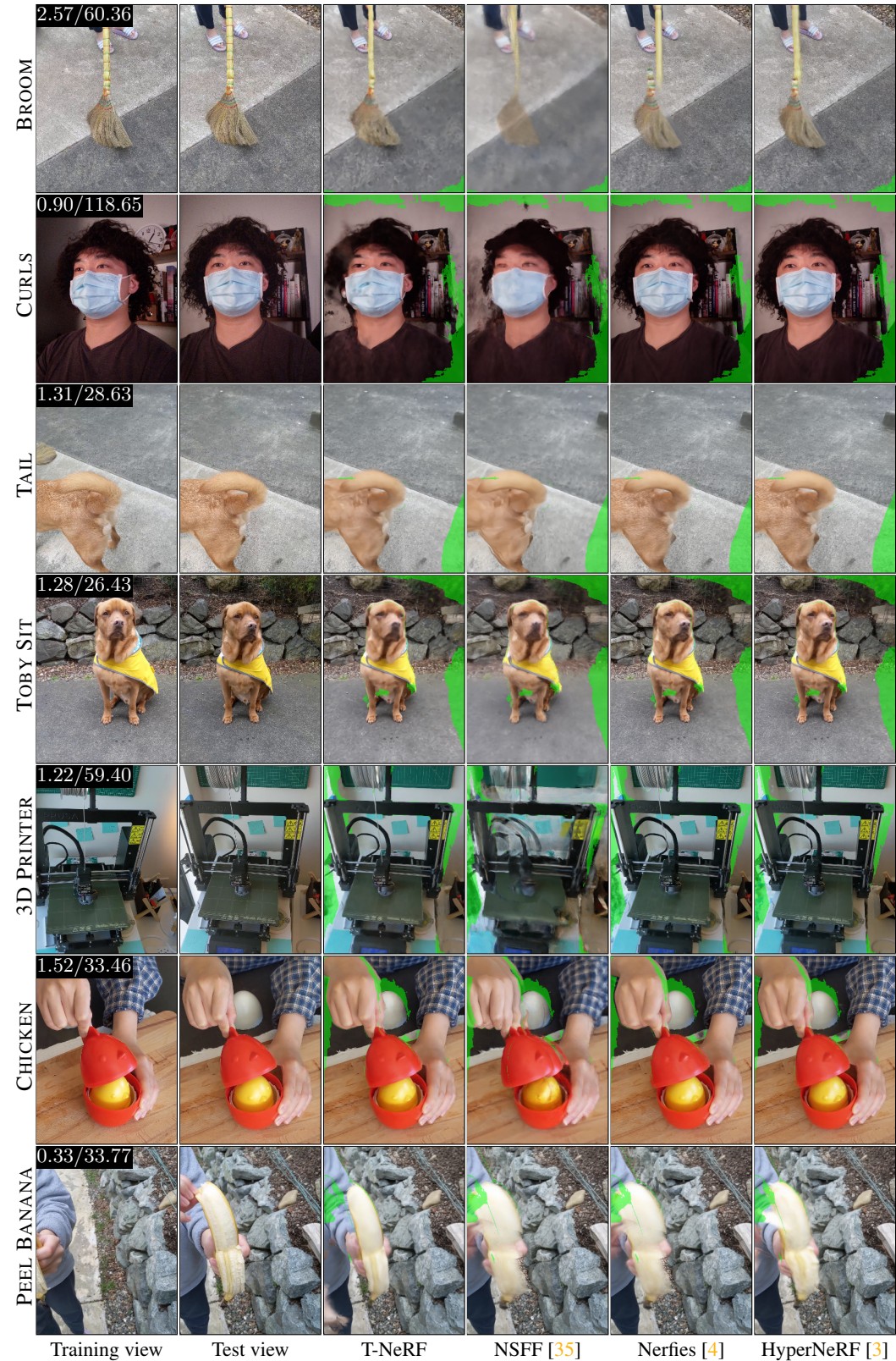

Figure 6: **Additional qualitative results on the Nerfies-HyperNeRF dataset without camera teleportation.** $\Omega/\omega$ metrics of the input sequence are shown on the top-left.

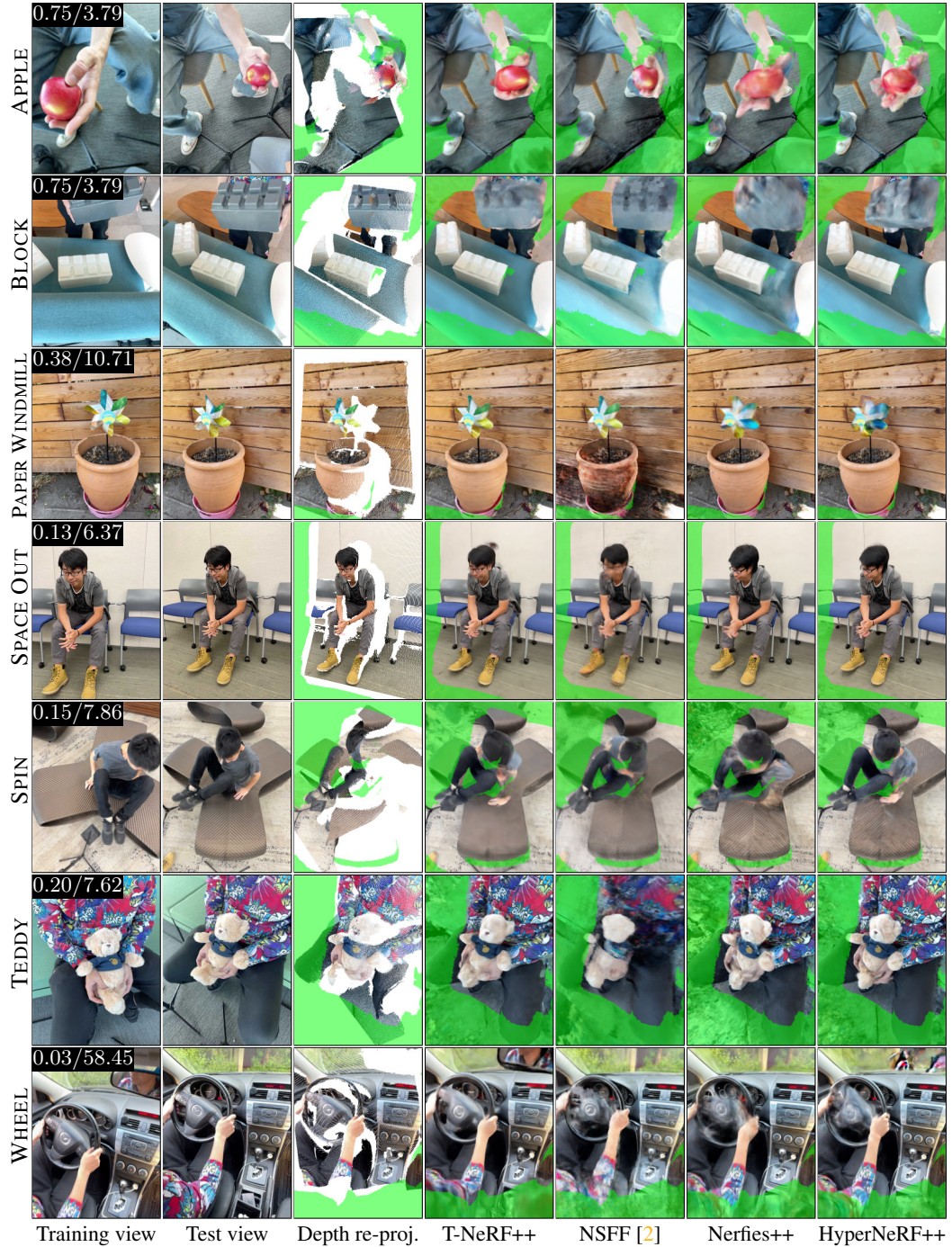

Figure 7: **Additional qualitative results on the multi-camera captures from the proposed iPhone dataset.** $\Omega/\omega$ metrics of the input sequence are shown on the top-left. The models shown here are trained with all the additional regularizations (+B+D+S) except NSFF.

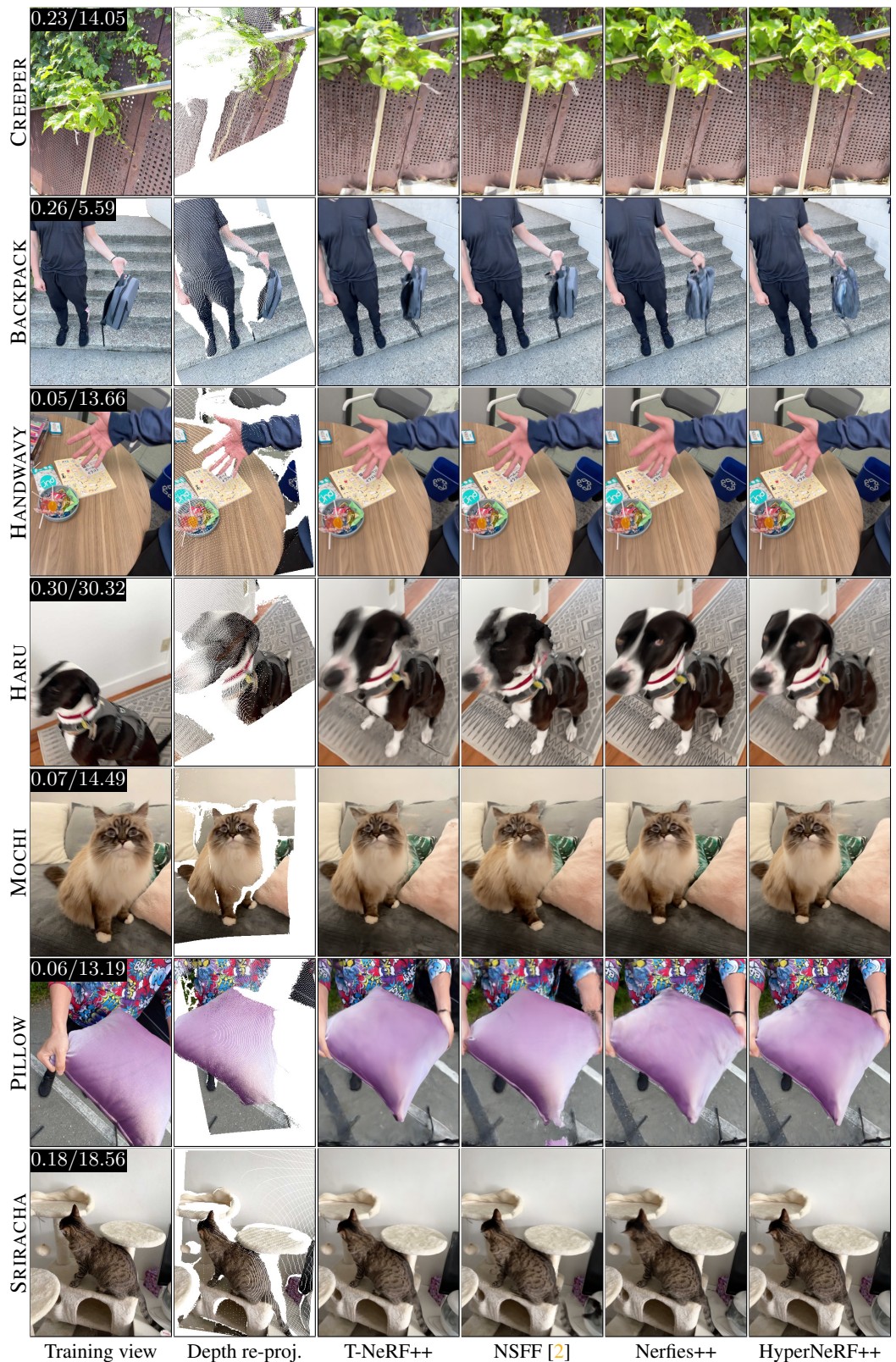

| Training view | Depth re-proj. | T-NeRF++ | NSFF [2] | Nerfies++ | HyperNeRF++ |

Figure 8: **Additional qualitative results on the single-camera captures from the proposed iPhone dataset.** $\Omega/\omega$ metrics of the input sequence are shown on the top-left. The models shown here are trained with all the additional regularizations (+B+D+S) except NSFF. We re-render the scene from the first viewpoint in each sequence. Note that there are no ground-truth validation frames.

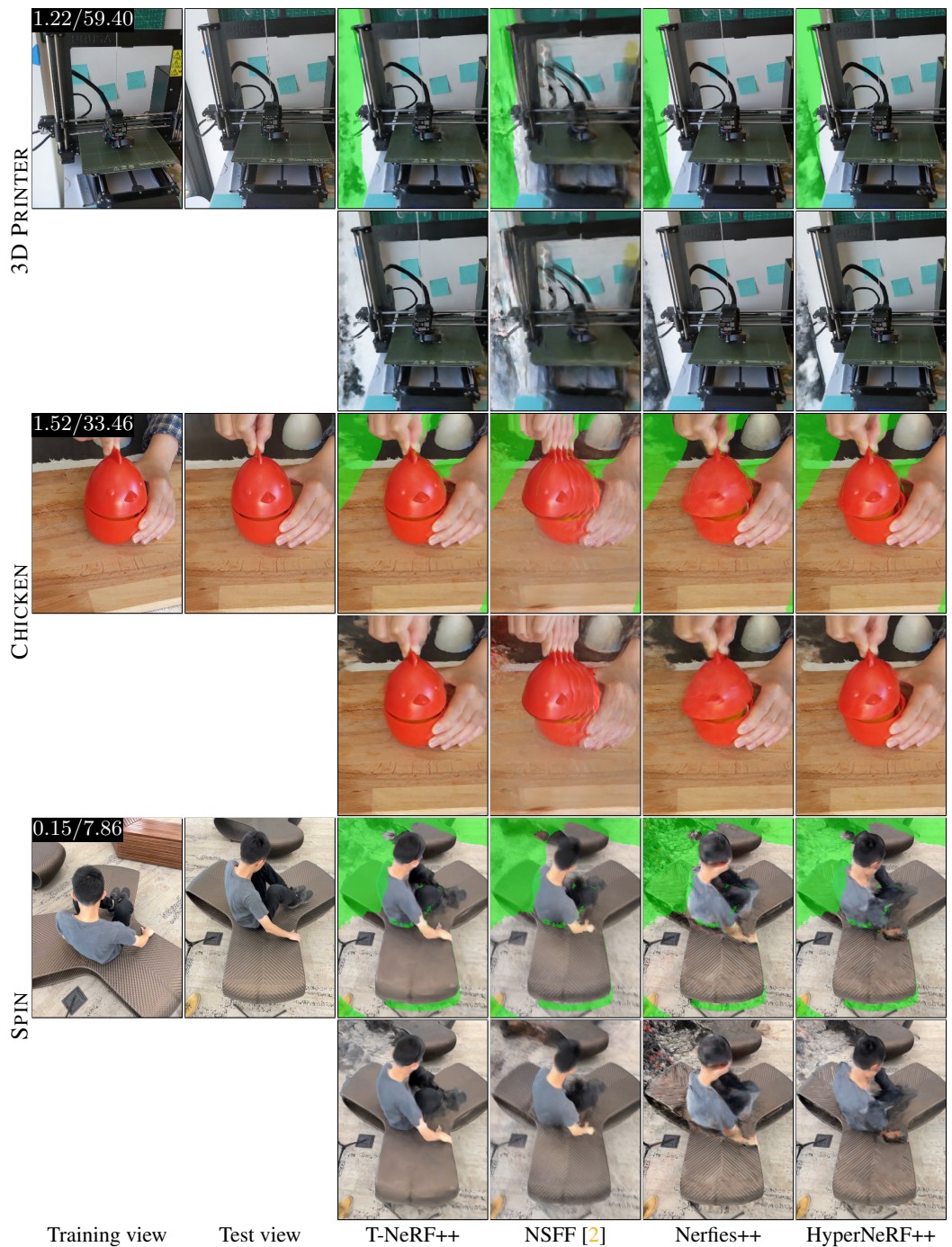

Figure 9: **Additional qualitative results on the full image rendering on both the Nerfies-HyperNeRF dataset and the proposed iPhone dataset.** $\Omega/\omega$ metrics of the input sequence are shown on the top-left. All models are trained under non-teleporting setting. For every two rows, we show the results with and without applying co-visibility mask. All models are not able to reconstruct the unseen regions.

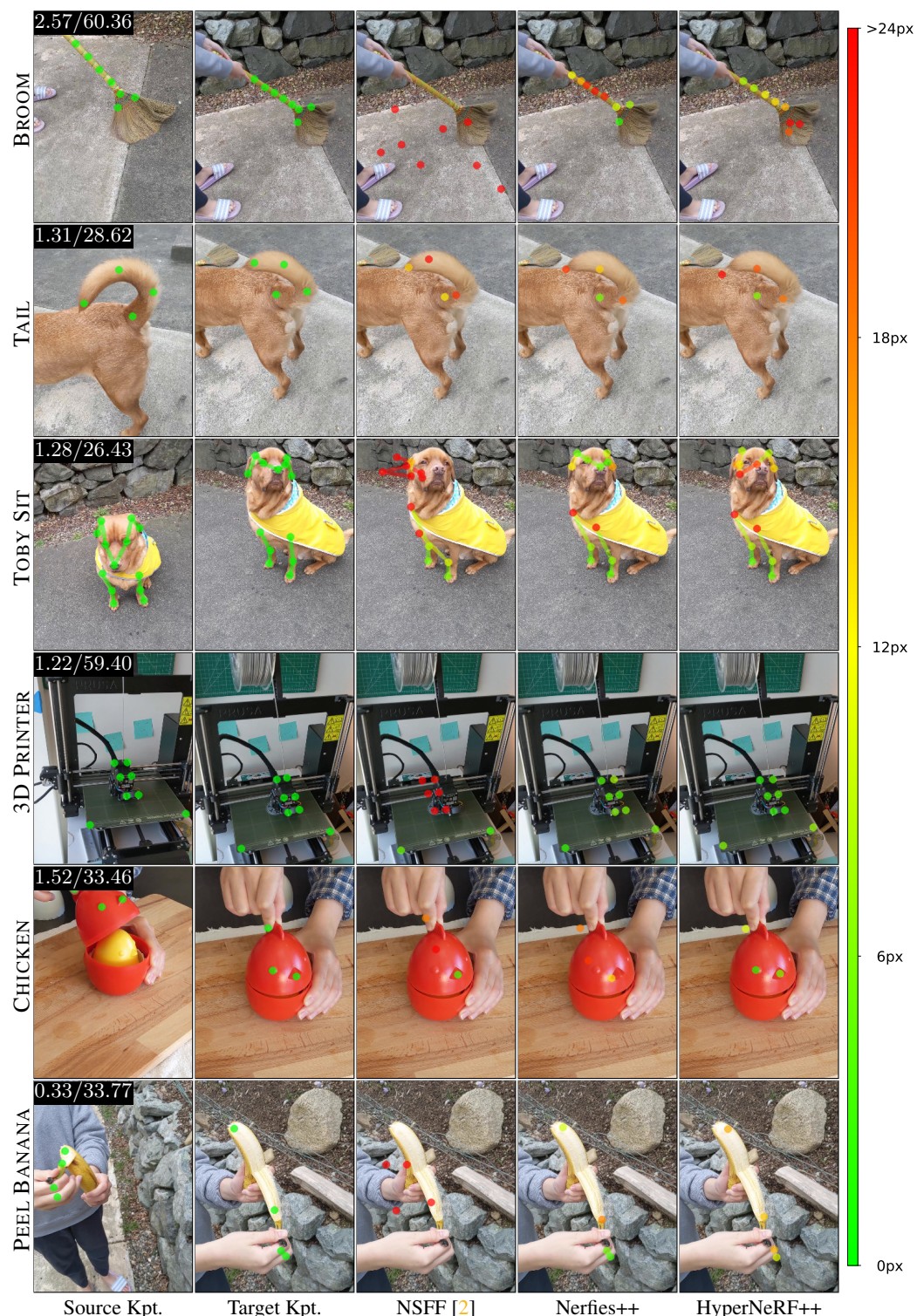

Figure 10: **Additional qualitative results of keypoint transferring on the Nerfies-HyperNeRF dataset without camera teleportation.** $\Omega/\omega$ metrics of the input sequence are shown on the top-left. All models are trained under non-teleporting setting. Transferred keypoints are colorized by a heatmap of end-point error, overlaid on the ground-truth target frame.

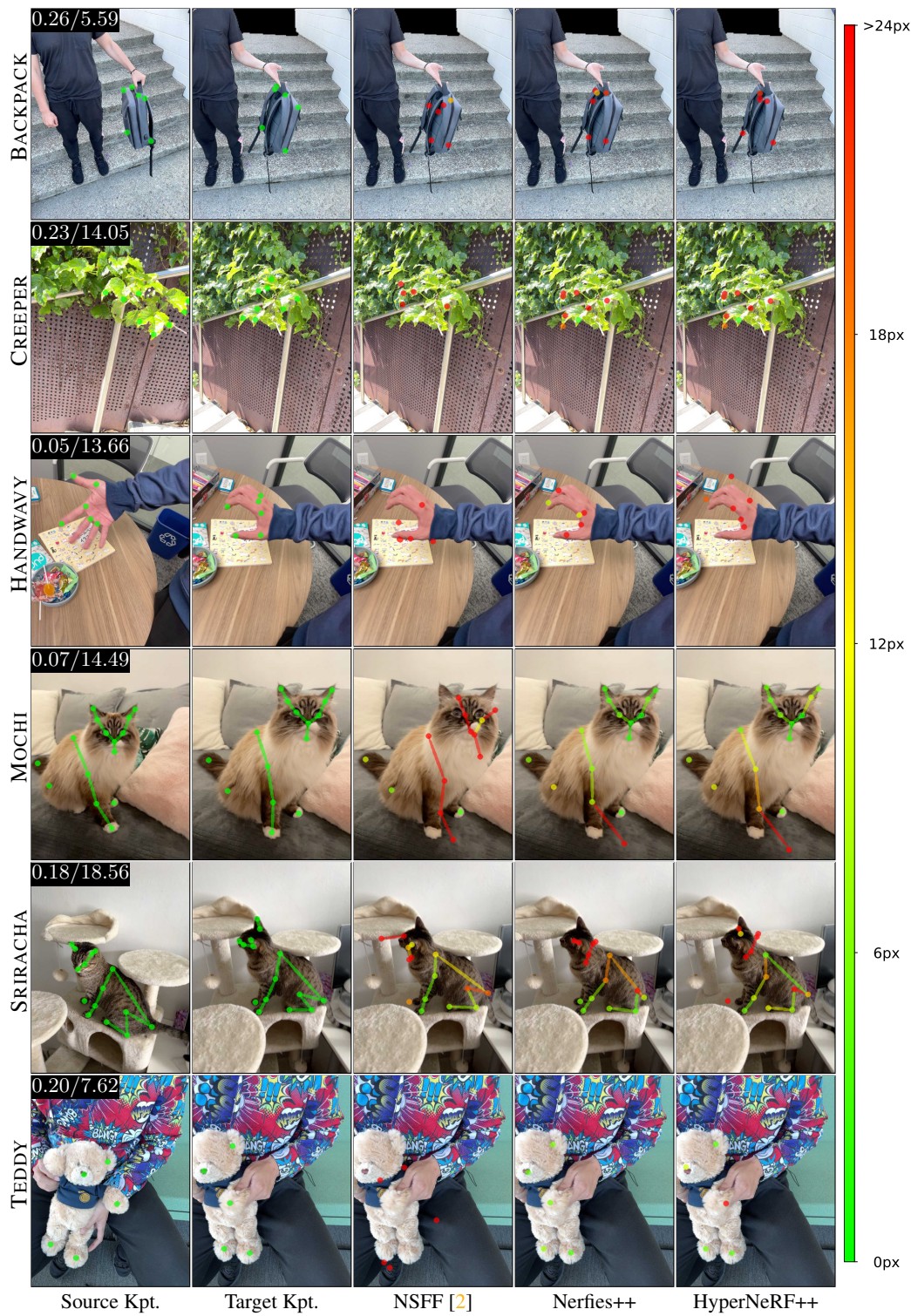

Figure 11: **Additional qualitative results of keypoint transferring on the proposed iPhone dataset.** $\Omega/\omega$ metrics of the input sequence are shown on the top-left. All models are trained under non-teleporting setting. Transferred keypoints are colorized by a heatmap of end-point error, overlaid on the ground-truth target frame.