# OpenReview forum: "Monocular Dynamic View Synthesis: A Reality Check"
_NeurIPS.cc/2022/Conference — NeurIPS 2022 Accept_

### Official Review · Reviewer_dLU6 · 2022-07-11

**Rating:** 6
**Confidence:** 4
**Soundness:** 3 good
**Presentation:** 3 good
**Contribution:** 3 good

**Summary:**

This paper proposes a dataset of monocular videos to evaluate non-rigid novel view synthesis methods. A factor for measuring the multi-view effect is proposed, and two evaluation metrics are proposed.

**Questions:**

If the authors propose a new method, it is good to evaluate it in the proposed datasets and evaluation metrics. However, from my perspective, the contribution is not significant enough for presentation in NeurIPS.

**Limitations:**

yes

**Strengths And Weaknesses:**

Strengths:

The paper is well-organized, and it is easy to read.

The datasets with evaluation metrics are proposed.

Several methods of non-rigid novel view synthesis are evaluated on the proposed datasets.



Weakness:

1. Using a multi-camera setup is more reliable than single-camera setup for performance analysis. After all, the goal is to evaluate methods instead of training models, I don’t think that it is the drawback of existing methods. We do not always need to capture new data for non-rigid novel view synthesis evaluation, so it is difficult for me to understand the strengths of the proposed solution.

2. The idea of Masked-PSNR is not surprising. First, if our goal is to evaluate the regions that are observed, it would be straightforward to mask other regions out. Second, we may also hope that the methods can predict and synthesize the regions that are not observed in training data. In this scenario, the mask should not be used. Overall, I don’t think that the masking is novel.

3. Although Nerfies trained models on images sampled from two cameras, the method is able to train models using a single-camera setup.

4. The factor of the multi-view effect is just a simple trick, and there exist many other variants if we want. It is correct, but I don’t think that it is sufficiently significant in this problem.

Post rebuttal,

After discussion, I agree with Reviewer vDbH, and I would strongly suggest the authors use the comments by vDbH in the final submission to position the work.

---

> ### Author Response · Authors · 2022-08-02
> **Response to Reviewer dLU6 [1/2]**
>
> We appreciate your thoughtful feedback. We respond to your comments below.
>
> **Meta point. “If the authors propose a new method, it is good to evaluate it in the proposed datasets and evaluation metrics. However …, not significant enough … for presentation in NeurIPS.”**
>
> Our contribution is an empirical analysis of the existing evaluation protocols for dynamic novel-view synthesis. Our study has found that the existing training and quantitative evaluation protocols for this task have been operating in an effectively multiview regime. Our study was inspired by impactful empirical studies for other tasks, such as object detection [1], object level single-view 3D reconstruction [2], and metric learning [3].  We hope that our work better calibrates the recent results on this task.
>
> 1. An empirical study of context in object detection. Divvala *et al.*, CVPR 2009.
> 2. What Do Single-view 3D Reconstruction Networks Learn? *Tatarchenko et al.*, CVPR 2019.
> 3. A Metric Learning Reality Check. *Musgrave et al.*, ECCV 2020.
>
> **Q1. “Using a multi-camera setup is more reliable than single-camera setup for performance analysis. After all, the goal is to evaluate methods instead of training models…We do not always need to capture new data.”**
>
> We agree that a multi-camera setup is more reliable than a single-camera setup for evaluating dynamic novel-view synthesis performance. Note that we also include a multi-camera evaluation in Table 2 of our paper.
>
> While we agree that the goal is to evaluate methods, we wish to point out that one of the main messages of our paper is that prior methods also effectively train in a multiview setting while claiming that they reconstruct from a “monocular”/single-view capture. Our contribution is not merely in proposing new data, but to expose this fact and propose a metric (EMF) in which future methods can quantify the difficulty of their training/evaluation setup based on the relative camera and object motion.
>
> To further motivate the need for capturing new data, please note that prior work trained and evaluated on data that often depicted objects that have slow motion. For example, please refer to the “peel-banana”, “tail”, “chicken”, and “toby-sit” sequences from the Nerfies/HyperNeRF captures on our supplemental webpage. This finding is one of our main motivations for capturing new data with larger and more complex motions in daily scenarios. As shown on the supplementary webpage, our “teddy” sequence depicts complex deformation of the toy body; our “block” and “wheel” sequences depict articulated motion. Our new captures augment existing benchmarks with larger object and smaller camera motions. We will add these points to the paper.
>
> Furthermore, we propose to evaluate correspondence by PCK-T and our dataset also comes with the labels to run this analysis, which has not been done before. Indeed when trained on real monocular capture sequences, these methods do not work as well. In the final version, we will also include the result with and without interleaving on existing datasets (our Table 1 for Reviewer vDbH-Q2 in this rebuttal), where the performance goes down without interleaving.
>
> **Q2. “The factor of the multiview effect is just a simple trick, and there exist many other variants if we want. … Not significant.”**
>
> We agree that our EMF formulation is simple, and allows quantifying that the existing training/evaluation has been operating in an effectively multiview regime. The novelty and importance of our work is in exposing issues in the current experimental protocol and pointing out the need for metrics like EMF. This observation has been overlooked by the community, and we hope that our findings can help instantiate better evaluation practices going forward.

---

> > ### Author Response · Authors · 2022-08-02
> > **Response to Reviewer dLU6 [2/2]**
> >
> > **Q3. “The idea of Masked-PSNR is not surprising… And we may also hope that the methods can predict … regions that are not observed.”**
> >
> > While the idea of Masked-PSNR is not surprising, prior approaches have not used this criterion before for evaluation. As a reminder, prior approaches have been evaluated by interleaving frames from different cameras to ensure that all testing regions have been observed during training. Quoting the Nerfies paper:
> > > “We alternate assigning the left view to the training set, and right to the validation, and vice versa. This avoids having regions of the scene that one camera has not seen.”
> >
> > However, as we have discussed, interleaving frames from different views during training leaks multiview cues, which no longer makes the evaluation monocular. Masked-PSNR goes around this issue so that no interleaving has to be done. As prior approaches have always quantitatively trained/evaluated with interleaving, our proposal to use Masked-PSNR for evaluation on this task is novel. We will incorporate this discussion in the paper.
> >
> > We agree that it is interesting for methods to evaluate prediction and synthesis of regions that are not observed during training. Here we include more results on our iPhone benchmark when testing with and without masking unseen parts during training. We train all models with depth and sparsity regularization.
> >
> > |           | PSNR with masking | PSNR without masking |
> > |-----------|:-----------------:|:--------------------:|
> > | T-NeRF    |        16.4       |          9.5         |
> > | Nerfies   |        16.9       |          9.1         |
> > | HyperNeRF |        16.7       |          9.3         |
> > | NSFF      |        14.0       |          9.2         |
> >
> > **Table 2.** Evaluating the state of the art on the proposed iPhone benchmark with and without masking unseen regions during training.
> >
> > Notice that the PSNR significantly degrades for the regions that are not observed during training. We will include these results and discussion in the experiment section of our paper.
> >
> > **Q4. “Although Nerfies trained models on images sampled from two cameras, the method is still able for a single-camera setup.”**
> >
> > It is true that Nerfies can train on a single-camera input. However, they do not report a quantitative evaluation for the single-camera setup without interleaving frames from the two views during training. Thus, we argue that their training/evaluation setup does not qualify as “monocular”.
> >
> > Our work is the first to show a “monocular” training/evaluation protocol. We have also benchmarked previous methods before and after fixing the interleaving issue during training using our evaluation protocol. We have observed that there is a huge performance gap in Table 1 (this rebuttal) to Reviewer vDbH-Q2, aligning well with what we have found on our iPhone benchmark.

---

> > > ### Comment · Reviewer_dLU6 · 2022-08-08
> > > **RE: Response to Reviewer dLU6**
> > >
> > > Thanks for the detailed reply. I can feel that the authors put a lot of effort into this paper, including paper writing, experiments, and rebuttal. Also, I agree with Reviewer A6gt that there is not some technical flaw.
> > >
> > > My main concern is about the contribution, as mentioned by the author "Our contribution is an empirical analysis of the existing evaluation protocols for dynamic novel-view synthesis. Our study has found that the existing training and quantitative evaluation protocols for this task have been operating in an effectively multiview regime."  More specifically, I do not feel that the authors really solve an important problem.
> > >
> > > (1) If the problem is that the previous method uses multi-view data in training. e.g., interleaving frames, we can just encourage the future work to train models from a monocular video only. We can be more careful when reviewing papers.
> > >
> > > (2) If the problem is that it is hard to evaluate methods on monocular videos, we can just use a multi-camera setup that is more reliable. I do not think that the multi-view setup is an issue in evaluation, although I agree that a monocular training method is more promising.
> > >
> > > I still need some time to make a decision.

---

> > > > ### Author Response · Authors · 2022-08-08
> > > > **Thanks for your comments**
> > > >
> > > > We would like to thank the reviewer for the discussion, and wanted to offer some thoughts on the issues raised.
> > > >
> > > > In particular, we wholeheartedly agree with the statement that “We can just encourage the future work to train models from a monocular video only. We can be more careful when reviewing papers''. Unfortunately, this does not ‘just’ happen — and papers like ours are precisely the mechanisms that encourage the community to do this. One of our key takeaways is that one should consider the ‘multi-viewness’ of the data under which a method is shown to work, and our hope is that this work provides both the motivation and an empirical means for the community to ‘be more careful’ regarding this aspect.
> > > >
> > > > On the other hand, we respectfully disagree with the claim that “If the problem is that it is hard to evaluate methods on monocular videos, we can just use a multi-camera setup that is more reliable”. We do agree that using a multi-camera system for validation only is “reliable”, as we have shown in Table 2 of our paper and Table 1 in this rebuttal. Effective as it is, in practice, setting up a reliable multi-camera system in-the-wild, however, is challenging due to time-synchronization, different exposure and camera calibration. Our paper responds to this issue by additionally evaluating correspondences over monocular training views. In Table 2 of our paper, we have shown that the correspondence evaluation results, in absence of multi-view validation data, strongly correlate with a model’s performance in dynamic novel-view synthesis. We hope our proposal expands the types of data our community can exploit for evaluating methods.

---

### Official Review · Reviewer_A6gt · 2022-07-11

**Rating:** 7
**Confidence:** 4
**Soundness:** 4 excellent
**Presentation:** 3 good
**Contribution:** 4 excellent

**Summary:**

The paper does a complete review for existing approaches recovering dynamic 3D scenes from monocular videos, especially Nerfies, HyperNeRF and NSFF. It studies the camera trajectory, proposes a metric Effective Multiview Factors (EMF) to quantify multi-view cues in a dynamic scene with moving cameras. Existing datasets typically have high multi-view cues. Therefore, a new dataset captured by iPhone is introduced with little multi-view cue. Additional metrics such as Masked PSNR and PCK-T are also introduced to ensure the fairness of evaluation. The new benchmark brings additional challenges for existing approaches of dynamic 3D capture.

**Questions:**

> (L213 - 215) Each training sequence is annotated with 10 to 20 keypoints in every 10 frames

How do you select these keypoints?


**Limitations:**

Limitations are not discussed in the paper.

**Strengths And Weaknesses:**

Strengths:

- Quantifying multi-view cues using EMF is neat. Different datasets in dynamic 3D capture proposes different data including different camera trajectories. And EMF quatifies the difficulties of all the data.
- I also like PCK-T correspondence besides normal PSNR. Right now a lot of novel view synthesis approaches focus on PSNR but PSNR does not directly reflect how well the model understands the 3D world. PCK-T is more explicit and I think reporting both masked PSNR and PCK-T is a good idea.
- Supp webpage is fantastic. Thank you!

Weaknesses:

I don't see any weaknesses, although I think the paper is more like a new benchmark -- It might be more suitable for the benchmark track.

Additional comments (no need to address):

> Page 8 footnote: We find that this code base performs better than the original code release

missing a period.

---

> ### Author Response · Authors · 2022-08-02
> **Response to Reviewer A6gt**
>
> Thanks for your feedback. We appreciate that you find the need for EMF to characterize the (multiview) difficulty of the data, and that you see the need for PCK-T correspondences for evaluating deformation based approaches. We will add a discussion on limitations in the final paper. Please see our responses below.
>
> **Q1. “How do you select keypoints for each training sequence?”**
>
> For sequences of human and quadrupeds (dogs or cats), we annotate keypoints based on the skeleton defined in the COCO challenge [1] and AnimalPose [2]. For sequences that focus on more general objects (like “block” or “teddy”), we manually identify and annotate 10 to 20 trackable points across frames. We will add these details in the final version.
>
> 1. Microsoft COCO: Common Objects in Context. Lin *et al.*, ECCV 2014.
> 2. Cross-Domain Adaptation for Animal Pose Estimation. Cao *et al.*, ICCV 2019.
>
> **Q2. “Might be better for a benchmark track.”**
>
> Thanks for your suggestion. While we provide a dataset with low multiview cues, the main goal of our work is to point out the issues (or the discrepancies between the claims and the evaluation protocol) of recent works in this domain in the spirit of papers like [1,2,3], where we believe the main track serves this purpose better.
>
> 1. An empirical study of context in object detection. *Divvala et al.*, CVPR 2009.
> 2. What Do Single-view 3D Reconstruction Networks Learn? *Tatarchenko et al.*, CVPR 2019.
> 3. A Metric Learning Reality Check. *Musgrave et al.*, ECCV 2020.
>
> **Q3. “Limitations are not discussed.”**
>
> We will add a discussion on limitations in the final paper. As also mentioned to Reviewer 2BcV-Q2, our work only addresses the effect of relative camera and object motion in evaluating dynamic novel-view synthesis from a monocular video. There are other factors that affect the difficulty of the task, such as scene appearance, lighting conditions, difference between training and testing views, and type of object/scene deformation. These factors are important and beyond the scope of our paper.

---

> > ### Comment · Reviewer_A6gt · 2022-08-07
> > **Re: Response to Reviewer A6gt**
> >
> > Thanks for the rebuttal. That answers my questions.
> >
> > I've also read other reviews. For reviewer 2BcV, I agree "real word video sequences could also contain a well proportion of high EMF data". However, as authors write, their goal is to expose that existing methods primarily train and evaluate on high EMF data. At the same time, personally I don't think it's reasonable to reject a paper simply because of this (without mentioning any other weaknesses).
> >
> > For reviewer dLU6, the primary concern is (1) the setup and (2) Masked-PSNR/EMF are simple. However, a simple approach does not mean it's ineffective. As long as it solves the problem, it's great.
> >
> > Especially, I don't think these weaknesses lead to "3: Reject" since it should have "technical flaws, weak evaluation, inadequate reproducibility and incompletely addressed ethical considerations." I don't find any reviewers points a technical flaw.
> >
> > Overall, I decide to keep my rating.

---

### Official Review · Reviewer_2BcV · 2022-07-12

**Rating:** 3
**Confidence:** 2
**Soundness:** 2 fair
**Presentation:** 2 fair
**Contribution:** 1 poor

**Summary:**

This paper studied the problem of dynamic 3D scene synthesis from monocular video sequence and found flaws of over-representation of slow-moving objects with a fast-moving camera in existing datasets. The authors then proposed a new metric called effective multi-view actors (EMF) to quantify the amount of multi-view signal in the image sequence. The authors also introduced a new dataset with very low EMF and argued that the new dataset should be more suitable for evaluation of dynamic 3D scene synthesis methods. Finally the authors evaluated four representative algorithms on the new dataset and find performance gap not being noticed with previous existing datasets.

**Questions:**

When evaluate on different sequences with different EMF value, do we always find lower loss for high EMF sequences on existing methods?

**Limitations:**

The authors answered YES for the question “Did you describe the limitations of your work” but not stating which section contains it.

**Strengths And Weaknesses:**

Strengths:
- The authors delve into the characteristic of existing dataset and managed to produce innovative metric to evaluate the difficulty of dataset in terms of monocular dynamics.

Weaknesses
- While the ability of build 3d representation from monocular dynamics is desirable, real word video sequences could also contain a well proportion of high EMF data. So the argument that low EMF datasets is better for evaluation may not hold unconditionally. The difficulty of dataset may also come from other aspects, such as shape complexity and surface property of objects.

---

> ### Author Response · Authors · 2022-08-02
> **Response to Reviewer 2BcV**
>
> Thanks for your comments. We are glad that you have found our metrics to be innovative. Please refer to our responses to your questions below.
>
> **Q1. “Real world video sequences could also contain … high EMF data. So the argument that low EMF datasets are better for evaluation may not hold unconditionally.”**
>
> We want to clarify that our message is not that low EMF captures are always better. Instead, our goal is to expose that existing methods primarily train and evaluate on high EMF data  (with interleaving cameras), while claiming “monocular capture”. This setup does not reflect everyday videos captured with a single hand-held camera (e.g., a smartphone). The contribution of our paper is to quantify the relative camera and object motion (“multiview-ness”) of a monocular capture, and to suggest future works to report the EMF on newly captured data, as supported by Reviewer vDbH.
>
> **Q2. “The difficulty of a dataset may also come from other aspects.”**
>
> Agreed. There are many other factors that might affect the difficulty of a capture. In our work, we mainly focus on the effect of camera and object motion and show it is a significant factor, which has not been revealed before. A non-exclusive list of other factors affecting the difficulty of a capture might include scene appearance, lighting conditions, difference between training and testing views, and type of object/scene deformation. We will add this discussion in the main paper.
>
> **Q3. “Do high EMF sequences always have better performance?”**
>
> We have reported a new experiment in Table 1 (this rebuttal) to Reviewer vDbH-Q2 where we disabled the interleaving cameras of the data and trained all methods with a single camera (the same sequence but with lower EMF). The performance drops for all methods by a large margin. This finding suggests that high EMF sequences will always have better performance, when all other aspects are fixed, e.g., scene appearance, lighting conditions, difference between training and testing views, and type of object/scene deformation.
>
> **Q4. “Limitation section.”**
>
> Thanks for your feedback. We will add a discussion about limitations in the final paper. Our EMF formulation only quantifies the impact of relative camera and object motion to the evaluation of dynamic novel-view synthesis from a monocular video. Other factors, like scene appearance and lighting conditions, can also affect the evaluation, which are also important and beyond the scope of our paper.

---

### Official Review · Reviewer_vDbH · 2022-07-14

**Rating:** 6
**Confidence:** 4
**Soundness:** 3 good
**Presentation:** 4 excellent
**Contribution:** 3 good

**Summary:**

This paper studies the effective metrics for evaluating recent works on novel view synthesis from monocular videos. The proposed effective multi-view factors (EMF) measures multi-view signals in the evaluation of view synthesis. It also comes with a new dataset which tries to mitigate the multi-view signals in captures. Two new metrics are proposed to measure the quality of view synthesis and motion. Several state-of-the-art methods are further improved, but still struggle in the cases of motion and few multi-view signal.

**Questions:**

See above

**Limitations:**

Limitations are not discussed. The author can discuss the use case of EMF and two new metrics more.

**Strengths And Weaknesses:**

# Strengths
- EMF

The effective multi-view factors measure the amount of ''multi-viewness'' in the capture. Such ''multi-viewness'' makes the task of novel view synthesis on dynamic scenes easier as the problem degrades into a multi-view setup. EMF consists of scene motion and camera angular velocity. And higher value in either suggests there is a strong multi-viewness in the capture. EMF basically tells how easy/difficult the capture is to reconstruct. Such metrics is nice to have in the future dataset release.

- Masked PSNR and Correspondence

Two new metrics are proposed to further measure the quality of the synthesized image. Both are tailored for dynamic scenes, but I see them also useful in static scenes as well. Masked PSNR only calculates PSNR in the valid region, and PCK-T tells whether the predicted motion is correct. The design choices are smart.

# Weaknesses

- Correspondence

I see masked PSNR can be applied to any method as long as the ground truth pose and depth are available. However, the correspondence metrics lacks generalizability. As shown in the paper, only methods that explicitly models motion can calculate the PCK-T score. It does not apply to methods like T-NeRF.

- Camera angular velocity

Nerfies and HyperNeRF have a high camera angular velocity in Tbl. 1. As far as I see, this is because it is switching frames between two cameras. If the captures are reordered in a way that uses frames from camera 1 first and camera 2 second (or only frames from camera 1), will Nerfies still work? If so, it will have a much smaller velocity, thus less EMF. It would be interesting to see how the method performs vs. EMF.

- No view direction or appearance encoding

It is mentioned in L297 both are turned off during training. But the goal of PSNR_M is to evaluate the quality of NVS in seen region. In practice, view direction and appearance encoding are essential in synthesizing new views. Turning them off will hurt the PSNR_M score a lot. Is there a specific reason doing so aside from overfitting?

- Typo
L32 agnitude -> magnitude

---

> ### Author Response · Authors · 2022-08-02
> **Response to Reviewer vDbH**
>
> Thanks for your feedback and appreciation. We are glad that you find our EMF is a nice metric to have in the future dataset release, and that you see our Masked PSNR and Correspondence as smart design choices for synthesized image evaluation. Below we will address your comments.
>
> **Q1. “Camera angular velocity is high due to switching between cameras. What if you only train with one camera, will Nerfies still work? It will have lower EMFs.”**
>
> Your intuition is correct that interleaving between cameras leads to high EMFs. We also experimented with training on a single camera with lower EMF using the seven sequences from Nerfies and HyperNeRF benchmarks. Table 1 (this rebuttal) shows that all methods suffer a significant performance drop (2dB in PSNR and 0.1 in LPIPS) when disabling interleaving. These results validate our argument that existing interleaving training schemes leak multiview cues. We plan to include this table as well as more visualization results into our next revision.
>
> |           | With interleaving: $\Omega$ = 2.18, $\omega$ = 166°/s | Without interleaving:  $\Omega$ = 0.79, $\omega$ = 21°/s  | Performance gap percentage |   |
> |-----------|:-----------------------------------------------------------:|:---------------------------------------------------:|:----------------------------:|---|
> |           |                        PSNR (dB) ↑, LPIPS↓                        |                    PSNR (dB) ↑, LPIPS↓                    | PSNR (dB) ↑, LPIPS↓              |   |
> | Nerfies   |                         22.2, 0.239                         |                     20.0, 0.352                     | -10%, +47%                 |   |
> | HyperNeRF |                         21.9, 0.241                         |                     19.7, 0.353                     | -11%, +46%                 |   |
> | NSFF      |                         25.2, 0.342                         |                     23.3, 0.512                     | -8.0%, +50%                  |   |
>
> **Table 1.** Evaluating the state of the art on existing Nerfies and HyperNeRF benchmarks with and without interleaving between frames during training.
>
> **Q2. “Correspondence metrics lack generalizability …only methods that explicitly model motion can calculate the PCK-T score. It does not apply to methods like T-NeRF. ”**
>
> We agree that the correspondence metric only applies to methods that explicitly reason about motion. Yet we want to argue that correspondence is an important aspect of non-rigid reconstruction [1,2] and is the foundation of many downstream applications such as content editing [3]. So we encourage future methods to explicitly model and evaluate it. As also pointed out by Reviewer A6gt, prior works focus on evaluating photorealism and overlooks the fact that photometric supervision may lead to erroneous correspondences (see Figures 1-3 in the supplemental materials). The correspondence accuracy will be captured by the PCK-T scores (see Table 2 in the main paper).
>
> 1. Optimal Step Nonrigid ICP Algorithms for Surface Registration. Amberg *et al.*, CVPR 2007.
> 2. DynamicFusion: Reconstruction and Tracking of Non-Rigid Scenes in Real-Time. Newcombe *et al.*, CVPR 2015.
> 3. TAVA: Template-free Animatable Volumetric Actors, Li *et al.*, ECCV 2022
>
> **Q3. “No view direction or appearance encoding.”**
>
> We turned them off since we noticed that models overfit and perform worse on our iPhone sequences. If requested, we’re happy to add results with the view-direction and appearance encoding enabled. For example, the models in Table 1 (this rebuttal) have both turned on, as per the default training recipes from the original code release. We find in practice that turning on the view-direction or appearance encoding does not affect our conclusions.
>
> **Q4. “Limitations are not discussed. The authors can discuss the use case of EMF and two new metrics more.”**
>
> Thanks for your suggestion; we will incorporate this feedback into our paper. Concretely, we will add a discussion that our EMF only addresses the effect of relative camera and object motion in evaluating novel-view synthesis from a monocular video, while in practice there are other factors like scene appearance and lighting conditions, which we leave for future works. We will also add a discussion that evaluating with Maksed PSNR and PCK-T requires scene depth acquisition and keypoint annotation. The former may not always be available and the latter requires human annotation. We hope to alleviate these issues by publicly releasing our tools for capturing and labeling.

---

> ### Comment · Reviewer_vDbH · 2022-08-09
> **Post rebuttal**
>
> I would like to thank the authors for the rebuttal. It addressed my concerns on the paper. I am in favor of an accept.
>
> The paper tackles an important problem -- measure the difficulty in dynamic novel view synthesis dataset. All four methods (Nerfie, HyperNeRF, T-NeRF, and NSFF) try to synthesize novel view from a dynamic capture. They all achieve SOTA numbers on their own dataset. However, it is hard to compare across different datasets -- some captures are easier to reconstruct than others. The proposed EMF metrics has shown a high correlation with the reconstruction quality. With the EMF score provided for each dataset, readers can have a better understanding how good the proposed method really is.
>
> Dynamic novel view synthesis is the task of both interpolation and hallucination. The method need to not only interpolate pixels from multi view cues but also hallucinate pixels for occlusion/dis-occlusion. The proposed EMF measures how much these NeRF-based methods benefits from multi view cues.
>
> I also agree reviewer A6gt's comments on other two reviews. Those two reject ratings are not supported by enough solid weakness arguments.

---

> > ### Comment · Reviewer_dLU6 · 2022-08-09
> > **Reply to Reviewer vDbH**
> >
> > Thanks for your comments. I think that your (vDbH) discussion about the "important problem" more directly addresses my main concern than the authors' reply. As there are no technical issues in this paper, as I mentioned before, I agree to accept the paper.
> >
> > I would suggest the authors highlight the discussion that you (vDbH) post here in the final submission.

---

> > > ### Author Response · Authors · 2022-08-09
> > > **RE: Post rebuttal**
> > >
> > > We’d like to thank the reviewers for the discussion. We are glad that they are now positive about the work, and would request the reviewer to kindly update their rating/review to reflect this. We will modify the text in the final version using the comments from Reviewer vDhB to help position the work and its importance.

---

### Author Response · Authors · 2022-08-02
**General response**

We would like to thank all the reviewers for their thoughtful comments. While we address the specific concerns raised in detail in individual comments, we would like to take this opportunity to highlight why we believe our paper would significantly benefit the community.

A growing number of recent works have addressed (neural) reconstruction of dynamics scenes. While we have seen several impressive algorithmic innovations in this area, we believe that the commonly prevalent evaluation practices have led to a disconnect between the empirically validated setups (effectively-multiview captures) and the claimed applications (e.g. reconstruction of generic scenes from casual monocular videos). We view our work as providing a first systematic means of characterizing this discrepancy (via the proposed EMF characterization) and highlighting the corresponding performance implications.

We of course agree that our work is not the final word on how one should analyze the difficulty of the learning setup, but strongly believe that our EMF characterization, along with suggested practices for evaluation in monocular setups as well as a representative dataset will serve as a starting point for the community. In particular, we hope that like papers [1,2,3] which helped move their respective fields forward, our work can help instantiate better experimental practices for future works in our field.

1. An empirical study of context in object detection. Divvala *et al.*, CVPR 2009.
2. What Do Single-view 3D Reconstruction Networks Learn? Tatarchenko *et al.*, CVPR 2019.
3. A Metric Learning Reality Check. *Musgrave et al.*, ECCV 2020.

---

### Meta-Review · Area_Chair_hMJp · 2022-08-22

**Recommendation:** Accept
**Confidence:** Certain

**Metareview:**

Pre-rebuttal, this paper had mixed reviews. Post-rebuttal, the paper had two strong supporters, A6gt and vDbH, who argued that the paper provides valuable insights into an important field, as well as a supporter dLU6, who commented in the discussion below that they are in favor of the paper (although did not update their review). The only remaining criticism comes from 2BcV. The AC does not find 2BcV's review persuasive (A6gt's comments summarize the AC's perspective well) and 2BcV did not participate in discussion. The AC is inclined to accept the paper and encourages the authors to use their extra page to integrate their responses to the reviewers.

**Award:**

No

---

### Decision · Program_Chairs · 2022-09-14

Accept